# Post-exercise hot water immersion enhances haemodynamic and vascular benefits of exercise without further improving cardiorespiratory fitness, glucose, lipids or inflammation

Charles J. Steward[1,2] ⓘ, Mathew Hill[1] ⓘ, Campbell Menzies[1,2] ⓘ, Sophie L. Russell[1,3] ⓘ, C. Douglas Thake[1] ⓘ, Christopher J. A. Pugh[4,5] ⓘ and Tom Cullen[1] ⓘ

[1]Centre for Physical Activity, Sport and Exercise Sciences, Coventry University, Coventry, UK
[2]The David Greenfield Human Physiology Unit, School of Life Sciences, Queen's Medical Centre, University of Nottingham, Nottingham, UK
[3]Centre for Nutrition, Exercise and Metabolism, Department for Health, University of Bath, Bath, UK
[4]Cardiff School of Sport & Health Sciences, Cardiff Metropolitan University, Cardiff, UK
[5]Centre for Cardiovascular Research, Innovation and Development, Cardiff Metropolitan University, Cardiff, UK

Handling Editors: Karyn Hamilton & Zachary Schlader

The peer review history is available in the Supporting Information section of this article (https://doi.org/10.1113/JP288873#support-information-section).

*The Journal of Physiology*

**Abstract figure legend** Eight weeks of 30-min moderate-intensity aerobic exercise followed by 40°C water immersion, on average three times per week, reduces diastolic and mean arterial blood pressure, and increases brachial artery flow-mediated dilatation more than post-exercise thermoneutral immersion at 34°C. These adaptations occurred in the absence of further enhancing cardiorespiratory fitness, glucose, lipids or inflammation in middle-aged adults not meeting the minimum physical activity guidelines. Abbreviations: post-exercise hot water immersion (EX+HWI), post-exercise thermoneutral water immersion (EX+TWI), cardiovascular disease (CVD), maximum heart rate (HRmax), body mass index (BMI), diastolic blood pressure (DBP), mean arterial pressure (MAP), flow-mediated dilatation (FMD). Created with Biorender.com.

The Journal of Physiology

**Abstract** There is considerable overlap between the mechanisms underlying the health benefits of exercise training and heat therapy. However, it remains unclear whether combining heat therapy with exercise can enhance improvements in cardiovascular and metabolic health. The present study investigated whether post-exercise hot water immersion (EX+HWI) could augment improvements in cardiorespiratory fitness, cardiovascular and metabolic health compared to post-exercise thermo-neutral water immersion (EX+TWI). Twenty-four physically inactive middle-aged adults (age: $58 \pm 5$ years; body mass index: $28 \pm 3$ kg m$^{-2}$; 13 females) were randomly allocated to 8 weeks of supervised EX+HWI ($n = 12$) or EX+TWI ($n = 12$). Moderate-intensity aerobic exercise (65–75% maximum heart rate) was performed for 30 min followed by 30 min of immersion at 40°C or 34°C, two to four times per week (total 24 sessions). Cardiorespiratory fitness, brachial artery flow-mediated dilatation, aortic pulse wave velocity, blood pressure and circulating lipids, glucose and inflammatory markers were assessed pre- and post-intervention. Between-group differences showed that EX+HWI resulted in greater reductions in mean arterial pressure ($P = 0.029$, $\eta^2_p = 0.207$, mean difference: –4 mmHg) and an increase in brachial artery flow-mediated dilatation ($P = 0.030$, $\eta^2_p = 0.206$, 2.33%). In addition, there were greater improvements in perceived physical health ($P = 0.036$, $\eta^2_p = 0.211$, 5 a.u.). No between-group differences were observed for cardio-respiratory fitness, aortic stiffness, circulating glucose, lipids and inflammatory markers. Taken together, post-exercise hot water immersion enhances blood pressure and brachial artery endothelial function, in the absence of improvements in cardiorespiratory fitness, circulating glucose, lipids and inflammatory markers.

(Received 14 March 2025; accepted after revision 1 July 2025; first published online 25 July 2025)
**Corresponding author** C. J. Steward: The David Greenfield Human Physiology Unit, School of Life Sciences, Queen's Medical Centre, University of Nottingham, Nottingham, NG7 2UH, UK. Email: charles.steward@nottingham.ac.uk

## Key points

- Hot water immersion replicates some of the physiological adaptations to exercise, but it is unclear whether post-exercise hot water immersion can augment exercise-derived improvements in cardiovascular health.
- This is the first study to assess whether 8 weeks of supervised exercise followed by hot water immersion can augment improvements in cardiovascular and metabolic health compared to post-exercise thermoneutral water immersion in physically inactive middle-aged adults.
- We demonstrate that 30 min of moderate-intensity aerobic exercise, followed by 40°C water immersion, on average three times a week for 8 weeks, reduces diastolic and mean arterial blood pressure more than exercise followed by thermoneutral immersion at 34°C, and increases brachial artery flow-mediated dilatation.
- These findings provide novel evidence that post-exercise hot water immersion lowers diastolic blood pressure and mean arterial pressure, and improves brachial artery endothelial function, without further enhancing cardiorespiratory fitness, glucose, lipids or inflammation in physically inactive middle-aged adults.

**Charles Steward** is a postdoctoral researcher at the University of Nottingham. He completed his PhD at Coventry University under the supervision of Dr Tom Cullen, with a research focus on the acute and chronic effects of post-exercise hot water immersion in physically inactive, middle-aged adults. Currently, Dr Steward works under the guidance of Professor Paul Greenhaff, co-ordinating and delivering research projects that utilise magnetic resonance imaging and stable isotope tracer methodologies. His ongoing work aims to advance the understanding of exercise-induced muscle damage and the cellular pathways involved in muscle function and metabolism following hip fracture surgery.

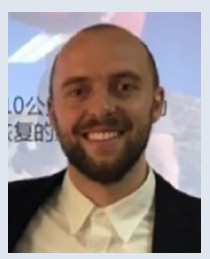

## Introduction

Cardiovascular disease (CVD) is the leading cause of mortality worldwide (WHO, 2023a). Remaining physically active throughout life reduces the risk of CVD by preventing the development of various modifiable cardiovascular and metabolic risk factors (Lloyd-Jones et al., 2006; Thijssen et al., 2019; Van Sloten et al., 2015). As these risk factors often begin to accumulate during middle-aged and often coincide with reductions in physical activity (Booth et al., 2011), it is crucial to intervene before irreversible cardiovascular damage occurs. Despite the irrefutable health benefits of exercise, many middle-aged adults do not meet the minimum recommended physical activity guidelines associated with reduced all-cause mortality (WHO, 2023b). However, some individuals do take part in infrequent, shorter and less intense bouts of physical activity in their weekly routine (e.g. brisk walking and cycling) (Luo & Lee, 2022). Because the transition from an inactive to a moderately active lifestyle elicits the largest reduction in all-cause mortality and CVD risk (Lee et al., 2014), initiating preventative lifestyle strategies that help mitigate declines in health are essential. It is therefore a priority to identify efficacious therapeutic approaches to use alongside smaller bouts of exercise to augment the cardiovascular and metabolic health benefits in physically inactive middle-aged adults.

Over the past decade, there has been significant research interest in the health benefits of heat therapy (Brunt & Minson, 2021). Epidemiological evidence suggests that higher frequencies of sauna use (Laukkanen et al., 2015) and hot water immersion (Ukai et al., 2020) are associated with reductions in the risk of cardiovascular events. Indeed, recent studies have begun to elucidate the physiological adaptations that may contribute to this reduction in CVD risk. For example, thrice weekly hot water immersion for 8–10 weeks has been shown to improve endothelial function, blood pressure, fasting glucose, total cholesterol and chronic low-grade inflammation (Brunt et al., 2016; Ely, Clayton, et al., 2019; Ely, Francisco, et al., 2019). Our research group has subsequently highlighted the considerable overlap between the mechanisms underpinning the health benefits of exercise training and heat therapy (Cullen et al., 2020). Indeed, exercise and heat therapy have both been shown independently to induce similar improvements in skeletal muscle capillarisation (Hesketh et al., 2019), endothelial function (Bailey et al., 2016) and cardiorespiratory fitness (Bailey et al., 2016; Hesketh et al., 2019). However, it is important to highlight that these positive findings occurred in young healthy adults and that there is currently limited evidence in populations at a higher risk of CVD. Given these overlapping mechanisms, it is conceivable that passive heating could be used in combination with exercise to further enhance an array of physiological adaptations derived from moderate-intensity exercise. In this regard, recent experimental work from our laboratory has demonstrated that a single session of post-exercise hot water immersion augmented exercise-derived elevations in conduit artery shear rate, heart rate, circulating nitrite and interleukin (IL)-6 (Steward et al., 2024), all of which are considered important acute stimuli for long-term cardiometabolic adaptations. As such, the findings from this study suggest that chronic post-exercise hot water immersion may induce greater improvements in cardiovascular and metabolic health compared to exercise training in isolation.

The available epidemiological evidence suggests that regular heat therapy, in combination with high cardiorespiratory fitness, confers additional health benefits and a greater reduction in the risk of all-cause mortality (Kunutsor et al., 2018). To date, the only randomised controlled trial investigating the additive health benefits of post-exercise heat therapy was conducted by Lee et al. (2022), who reported greater improvements in cardiorespiratory fitness, systolic blood pressure and total cholesterol after 8 weeks of post-exercise sauna use in physically inactive middle-aged adults. However, it remains unclear whether other forms of heat therapy can produce similar benefits. Moreover, although Lee et al. (2022) assessed several clinically relevant outcomes, they did not measure vascular function, glycaemic control, lipid profiles and markers of chronic low-grade inflammation. Accordingly, in the present study, we conducted a randomised controlled trial to assess the benefits of post-exercise hot water immersion on a wider range of clinically relevant outcomes of cardiovascular and metabolic health, which are important early predictors of CVD in middle-aged adults (Coutinho et al., 1999; Hedayatnia et al., 2020; Thijssen et al., 2019; Tuomisto et al., 2006).

We aimed to explore whether 8 weeks of post-exercise hot water immersion would improve cardiorespiratory fitness, vascular health and metabolic health by a greater extent than post-exercise thermoneutral water immersion. To do this, we measured an extensive array of health parameters, including maximal oxygen uptake, blood pressure, conduit artery endothelial function and stiffness, and circulating glucose, lipid and inflammatory markers. Using this approach, we sought to provide novel and important insights into the processes by which post-exercise hot water immersion may improve health outcomes associated with CVD in a population of physically inactive middle-aged adults. We hypothesised that hot water immersion after exercise would elicit greater improvements in vascular outcomes, cardiorespiratory fitness and circulating markers of cardio-

metabolic health compared to post-exercise thermo-neutral water immersion.

## Methods

### Ethical approval

Ethical approval was provided by Coventry University (P132234) and the study was pre-registered at ClinicalTrials.gov (NCT05409404). Prior to commencing the study, all participants completed informed consent forms and health questionnaires. This trial was reported in accordance with the Consolidated Standards of Reporting Trials guidelines (CONSORT) (Schulz et al., 2010). All procedures were conducted in accordance with the *Declaration of Helsinki*. Participant recruitment and testing took place from June 2022 to October 2024.

### Participants

One hundred and twelve participants were assessed for eligibility for this study. Of these individuals, thirteen physically inactive middle-aged females and eleven males met the criteria and completed the study (age: $58 \pm 5$ years; weight: $78 \pm 12$ kg; height: $168 \pm 8$ cm; body mass index (BMI): $28 \pm 3$ kg m$^{-2}$; $\dot{V}_{O_2peak}$: $18 \pm 3$ mL$^{-1}$ kg$^{-1}$ min$^{-1}$). Baseline anthropometrics and medications of each group can be found in the results section. The inclusion criteria for this study required participants to be aged between 45 and 60 years and classified as physically inactive, defined as engaging in <150 min of moderate-intensity or <75 min of vigorous-intensity activity per week, over the 6 month period prior to the study. Only non-smokers and individuals not taking anti-inflammatory medications were recruited. Self-confirmed postmenopausal women (i.e. >12 months without menstruation), who were not on hormone replacement therapy, were recruited to minimise the effect on circulating hormones on cardiovascular responses (Minson et al., 2000). Participants were excluded if they regularly used heat therapy (e.g. sauna and hot water immersion) or had travelled to a warm country in the 4 weeks before commencing the study. Participants were also excluded if they had a history of orthostatic hypotension, as passive heating can exacerbate presyncopal symptoms (Steward et al., 2023) or if they had cardiovascular, metabolic or renal disease.

### Experimental design

The study was a randomised between-groups design [block randomisation using the '*RAND*' function in Excel (Microsoft Corp., Redmond, WA, USA)] with participants allocated to either 8 weeks of supervised aerobic exercise training followed by hot water immersion (EX+HWI) or supervised aerobic exercise training followed by thermo-neutral water immersion (EX+TWI). Randomisation was conducted by the principal investigator prior to the pre-intervention health assessments. Participants were not informed that the EX+HWI group was the 'active' arm of the intervention, whereas the EX+TWI group served as the 'sham' group. As the principal investigator had to monitor water temperature and be aware of the risk of presyncopal symptoms (i.e. dizziness and nausea) when exiting the hot tub, the principal investigator was not blinded to group allocation. Health assessments were conducted pre- and post-intervention in the order provided in the schematic (Fig. 1) and in the health assessments section below.

An *a priori* sample size calculation ($\alpha = 0.05$ and $\beta = 0.80$) determined a total of 24 participants (12 in each group) would be required to detect an effect size of $d = 0.3$ using a repeated measures design (within-between interaction), with two groups and two measurement time points for our primary outcome measures of systolic blood pressure (SBP), diastolic blood pressure (DBP) and mean arterial blood pressure (MAP). This effect size was chosen based on the study of Akerman et al. (2019), who reported meaningful absolute changes, which equated to moderate to large effect sizes for reductions in SBP ($-4$ mmHg, $d = 0.57$), DBP ($-3$ mmHg, $d = 0.43$) and MAP ($-3$ mmHg, $d = 0.50$) following 12 weeks of pre-exercise heat therapy compared to exercise alone. This is supported by previous research indicating that lowering DBP by 2 mmHg in middle-aged men and women with a DBP >90 mmHg reduces the risk of coronary heart disease by 6% and stroke by 15% (Cook, 1995). Additionally, reducing SBP by 5 mmHg lowers the risk of future CVD events by 10% in both men and women around the age of 65 years, regardless of their initial blood pressure levels (Rahimi et al., 2021). Taken together, based on the study by Akerman et al. (2019), we chose to use a more conservative estimate ($d = 0.3$) of their observed blood pressure changes given the different populations being studied and protocols being used.

Alongside the primary outcome measures of SBP, DBP and MAP, the present study also assessed secondary outcome measures, including brachial artery flow-mediated dilatation, carotid to femoral pulse wave velocity and cardiorespiratory fitness. Additionally, exploratory outcome measures included glucose tolerance, venous lipids, inflammatory, angiogenic and vasoactive markers, future cardiovascular disease risk and quality of life.

Health assessments were conducted at the same time of day ($\pm 1$ h) to control for the potential influence of circadian rhythms and under ambient conditions (room temperature: $21 \pm 2$°C and relative humidity: $54 \pm 9$%) before and after the 8 week intervention. All post-intervention health assessments were conducted at

least 48 h after the most recent post-exercise immersion visit to avoid the residual acute responses from the stimulus. All assessments took place following an overnight fast and participants were instructed to avoid strenuous exercise, antihistamines, caffeine, alcohol and nitrite/nitrate rich foods (e.g. beetroot, rocket, spinach, lettuce cabbage radishes and highly processed meats) for 24 h before each visit. Participants taking medications did not alter type or dosage throughout the course of the study.

### Intervention characteristics

Throughout the 8 week supervised intervention, participants reported to the laboratory at Coventry University on average three times per week, either individually or in small groups (<4 participants). If a participant was unable to attend a training session, an additional session was completed the following week, therefore laboratory visits ranged from a minimum of two to a maximum of four sessions per week. This amounted to 24 training visits consisting of aerobic exercise training and water immersion within the required 8 week period.

### Aerobic exercise training

Supervised exercise sessions consisted of 30 min of aerobic exercise, divided into three consecutive 10 min blocks of rowing (Indoor Rower; Concept2, Morrisville, VT, USA), cycling (Wattbike, Nottingham, UK) and treadmill-based exercise (Desmo Treadmill; Woodway, Waukesha, WI, USA) in a randomised order. A variety of exercise modes were selected because this approach has been shown to improve exercise adherence in contrast to a single mode of exercise (Glaros & Janelle 2001). During the first 4 weeks, participants were instructed to maintain a

heart rate of ~65% of maximum heart rate. To facilitate exercise progression, the exercise intensity then increased to 75% of maximum heart rate for the final 4 weeks of the intervention. During exercise, heart rate was continually monitored by researchers and participants via heart rate monitors (FT1 Polar; Polar Electro Oy, Espoo, Finland) to ensure participants were within a zone that was $\pm 5$ beats min$^{-1}$ of the target heart rate. Heart rate training zones were based on each participant's maximal heart rate achieved during a maximal cardiopulmonary exercise test on a bike ergometer.

### Water immersion

Immediately following exercise, both groups took part in 30 min of supervised water immersion. In the EX+HWI group, immersion was at a temperature of $40.0 \pm 0.5$°C and EX+TWI at $34.0 \pm 0.5$°C. The duration and intensity/temperature of EX+HWI was selected based on our earlier work showing beneficial acute responses to a similar stimulus (i.e. core temperature, heart rate, vascular shear rate, IL-6 and nitrite) (Steward et al., 2024). Furthermore, as recommended by Brunt and Minson (2021), we included a thermoneutral water immersion group because there is evidence that hydrostatic pressure may independently contribute to acute cardiovascular responses (Carter et al., 2017; Löllgen et al., 1981), which may lead to some long-term adaptations (Simmons et al., 2017). Therefore, by using immersion rather than a non-immersed ambient control, we were able to isolate the benefits from passive heating *per se*. Immersion depth was kept constant up to the mid-sternum for all sessions. A temperature of 34°C was selected as thermoneutral in accordance with a recent review showing no acute change in rectal temperature, at the same time as

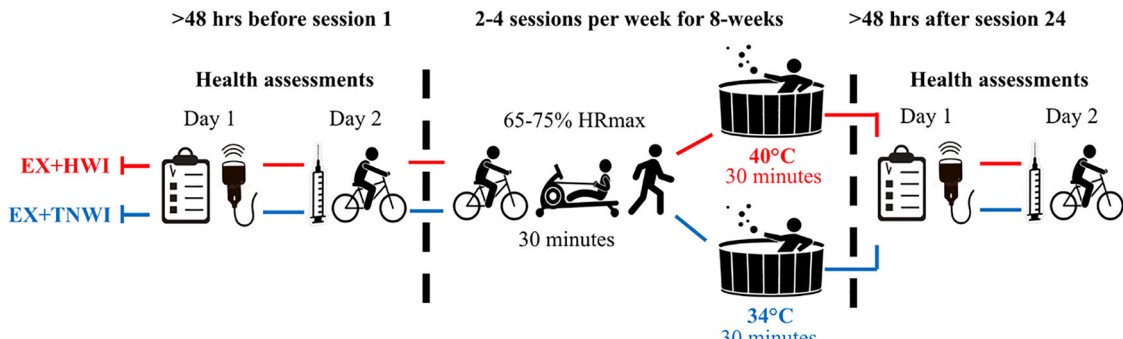

**Figure 1. A schematic representing the 8 week intervention and health assessments before and after for the EX+HWI and EX+TWI groups**

Participants were randomly allocated to 8 weeks of supervised EX+HWI (n = 12) or EX+TWI (n = 12). Moderate-intensity aerobic exercise (65–75% HRmax) was performed for 30 minutes, followed by 30 minutes of immersion at either 40°C or 34°C, 2–4 times per week (total of 24 sessions). Pre- and post-intervention health assessment visits were completed over two separate days, assessing cardiorespiratory fitness, brachial artery flow-mediated dilation, aortic pulse wave velocity, blood pressure, circulating lipids, glucose, inflammatory markers, and health-related quality of life. [Colour figure can be viewed at wileyonlinelibrary.com]

achieving a perceived neutral zone of thermal sensation (Ntoumani et al., 2023). The same commercially available hot tub (Lay-Z-Spa Majorca, Newton Abbot, UK) was used throughout the study. Additional hot/cold water was added or removed from the hot tub as required to maintain the water temperature and level. During the first 4 weeks, participants kept their arms out of the water and, in the final 4 weeks, arms were submerged to ensure progression of the heating stimulus (Steward et al., 2023).

## Health assessments

**36-Item Short Form Health Survey.** The 36-Item Short Form Health Survey assessed eight domains of self-reported health status, including physical functioning, physical role limitations, bodily pain, general health, vitality, social functioning, role emotional and mental health. The scores from each of the Likert scales were then converted to numerical values (0 worst possible health to 100 best possible health) (Ware, 2000). The domains' physical functioning, physical role limitations, bodily pain, vitality and general health were averaged to calculate the physical health component. The domains' role emotional, social functioning, mental health, vitality and general health were averaged to calculate the mental health component. These data were used to characterise health-related quality of life and can be found in the Supporting information (Table S1).

## Resting blood pressure

After 30 min of supine rest, blood pressure and heart rate were measured in duplicate. A 2 min rest period was provided between individual measurements of blood pressure (M3; Omron, Kyoto, Japan) and heart rate (FT1 Polar; Polar Electro Oy). A third measurement was taken for blood pressure if the first two SBP measurements had a difference >10 mmHg. The SBP, DBP and heart rate readings are reported as an average of the final two measurements. Pulse pressure (PP) was calculated as PP = DBP – SBP and MAP was calculated as MAP = DBP + 1/3 (PP). Before the intervention the test-retest coefficients of resting blood pressure were: SBP $4 \pm 2\%$, DBP $3 \pm 2\%$, PP $7 \pm 3\%$ and MAP $3 \pm 2\%$.

## Arterial stiffness

In line with expert consensus, carotid to femoral pulse wave velocity assessments were measured in duplicate after 10 min of supine rest (Townsend et al., 2015). All measurements were completed using a SphygmoCor Xcel unit (AtCor Medical, Naperville, IL, USA) with the inflatable cuff placed around the upper thigh of the

right leg. The distances were then recorded using a tape measure from the carotid pulse to the sternal notch, sternal notch to the top of the cuff and bottom of the cuff to femoral pulse for each participant. Pulse wave velocity measurements were taken by using a high-fidelity micromanometer-tipped probe. The probe was placed on the carotid pulse at the site of the maximal arterial pulsation. Upon detecting a concurrent signal from the tonometer and inflatable cuff, simultaneous femoral waveforms were captured for a minimum of 10 s, which allowed the calculation of aortic pulse wave velocity. If the measurements differed by greater than 0.5 m s$^{-1}$, a third measurement was taken (Townsend et al., 2015). The test-retest baseline coefficient across the two groups before the intervention for aortic pulse wave velocity was $3 \pm 2\%$. In addition, ultrasound-derived measures of beta stiffness index, dynamic compliance and conductance were calculated for the brachial artery, using:

$$\beta - \text{stiffness index} = \ln (SBP/DBP) / [(Ds - Dd)/Dd]$$

$$\text{(a.u.)}$$

$$\text{Dynamic arterial compliance} = [(Ds - Dd/Dd)/2\Delta PP]$$

$$*\pi Dd^2 \, (\text{mm}^2 \text{mmHg})$$

$$\text{Brachial artery conductance} = \text{bloodflow/MAP}$$

$$\left(\text{mmHg mL}^{-1} \text{ min}^{-1}\right)$$

where ln refers to the natural logarithm function, PP relates to pulse pressure and *Ds* and *Dd* relate to systolic and diastolic diameter, respectively.

## Endothelial function

Endothelial function was assessed via brachial artery flow-mediated dilatation (FMD) in line with current expert consensus guidelines (Thijssen et al., 2019), following at least 10 min of supine rest. The FMD assessment consisted of a 9 min ultrasound recording of the brachial artery in the distal third of the upper right arm. This consisted of 1 min recorded at baseline, followed by a 5 min period of forearm occlusion achieved via rapid cuff inflation >220 mmHg immediately distal to the olecranon process (Hokanson E20; Hokanson Vascular, Bellevue, WA, USA), and continued for 3 min after cuff release. Brachial artery diameter and velocities were recorded continuously and concurrently using pulse-wave mode and B-mode imaging with a 15 MHz multifrequency linear array probe attached to a high-resolution duplex ultrasound machine (Terason uSmart 3300; Teratech, Burlington, MA, USA). Images were taken following optimisation of the longitudinal

B-mode image of the lumen-arterial interface, with simultaneous Doppler velocities collected using the smallest possible insonation angle (always <60°). The same experienced sonographer performed all scans, and probe position (∼2 cm distal to antecubital fossa) and ultrasound settings were standardised for each individual participant and then replicated for the post-intervention measures.

Analysis of artery diameter and flow was performed using custom-designed edge-detection and wall-tracking software, which is independent of investigator bias and has been previously described elsewhere (Woodman et al., 2001). Reproducibility of diameter measurements using this semi-automated software is significantly better than manual methods, reduces observer error and possesses superior intra-observer reliability (Woodman et al., 2001). The lead researcher anonymised and coded all images, which were then analysed by a separate and experienced member of the research team who was blinded to the experimental group allocations. From synchronised diameter and velocity data, mean blood flow (the product of lumen cross-sectional area and Doppler velocity) was calculated at 30 Hz and shear rate, an estimate of shear stress without viscosity, was calculated as 4 × mean blood velocity/vessel diameter. Relative percentage change in FMD (FMD%) was calculated as (peak diameter – baseline diameter)/baseline diameter × 100.

### Oral glucose tolerance test

Fasted capillary blood samples were first taken from the fingertip. After which, participants ingested 75 g of glucose (Dextrose monohydrate; Myprotein, Northwich, UK) dissolved in 300 mL of water. Participants then remained seated for 120 min and fingertip blood capillary samples were then taken at 15, 30, 45, 60, 90 and 120 min. All capillary samples were taken in duplicate. Capillary whole blood samples were then analysed (Biosen C-Line Clinic; EKF-diagnostic GmbH, Barleben, Germany) and the average of the two samples for each time point was recorded. Data were reported as fasted glucose and area under the curve.

### Cardiorespiratory fitness

The cardiopulmonary exercise test was a ramp protocol on an electronically braked cycle ergometer (Lode Corival, Groningen, The Netherlands), which initially consisted of 2 min of unloaded cycling at 0 W, followed by an increase of 15 W min$^{-1}$ until voluntary exhaustion. For all cardiopulmonary exercise tests, the saddle height was adjusted accordingly before the pre-intervention test and replicated post-intervention. Participants were instructed

to maintain a cadence of 70–80 rpm. Expired gases were continuously measured by a breath-by-breath gas analyser (Ultima Series CPX; Medgraphics, Tewkesbury, UK) and heart rate through a strap connected to the Medgraphics software. An overall rating of perceived exertion based on peripheral (active muscles) and cardiovascular (heart and lungs) factors was measured every 2 min throughout the ramp test (Borg et al., 1987). Maximum heart rate achieved in the test was used to prescribe the 65% and 75% training heart rates during the intervention. Furthermore, ventilatory threshold 1 (VT$_1$) and ventilatory threshold (VT$_2$) were measured through the Medgraphics automated iterative regression and analysis of the slope (V-slope method). Accordingly, VT$_1$ was identified as the point at which carbon dioxide production increased faster than the increase in $\dot{V}_{O_2}$ and VT$_2$ was the point at which the increase in minute ventilation becomes faster than the increase in carbon dioxide production. Peak oxygen uptake ($\dot{V}_{O_2peak}$) was calculated as the average oxygen uptake over the final 30 s of cycling prior to voluntary exhaustion.

### Venous blood sampling and analysis

Fasted venous blood samples were obtained through venepuncture to the median cubital vein by a trained phlebotomist using 10 mL BD serum vacutainers (Nu-Care, Stewartby, UK). After obtaining the sample, serum samples were left to clot for 45 min and subsequently centrifuged for 12 min at 3000 **g**. Serum samples were then aliquoted into individual Eppendorfs and stored in a freezer at –80°C until analysis. Custom-designed Luminex multiplex assays were used to measure vascular endothelial growth factor (VEGF), IL-6, IL-10, IL-1$\beta$ and tumour necrosis factor alpha (TNF-$\alpha$) (Human HS Cytokine Premixed Performance Assay; R&D Systems, Minneapolis, USA). Commercially available enzyme-linked immunosorbent assay kits were used to measure endothelin-1 (ET-1) (DET100; Bio-Techne, Minneapolis, MN, USA), matrix metalloproteinase-9 (MMP-9) (DMP900, Bio-Techne) and cortisol (KGE008B; Bio-Techne). The intra-assay coefficients of variation for all assays were: ET-1: 5.2 ± 1.2%, MMP-9: 4.9 ± 0.3%, VEGF: 5.9 ± 0.5%, IL-6: 6.5 ± 1.5%, IL-10: 9.2 ± 0.1%, IL-1$\beta$: 8.7 ± 1.1%, TNF-$\alpha$: 5.9 ± 0.5% and cortisol: 6.2 ± 5.5%. IL-4 was also measured through the Luminex Multiplex assay; however, 85% of samples were below the detectable limit and thus not reported. A Randox Daytona+ (Randox Laboratories Ltd, Crumlin, UK) was used to analyse circulating lipid concentrations and C-reactive protein (CRP). All samples were analysed in duplicate, and assays were conducted in accordance with the manufacturer's instructions.

## Framingham 10-year cardiovascular risk score

The Framingham risk score was calculated to provide an overall composite risk level for the development of CVD over the next 10 years. The score was calculated based on a combination of established CVD risk factors, including sex, age, SBP, treatment for hypertension, smoking, diabetes, high-density lipoprotein and total cholesterol (D'Agostino et al., 2008).

## Characterisation of the heating stimulus

Resting tympanic temperature (Braun ThermoScan, Kronberg, Germany), heart rate (FT1 Polar; Polar Electro Oy) and perceptual measures, including thermal sensation (+5 hot to –5 cold), thermal comfort (+5 very comfortable to –5 very uncomfortable) (Gagge et al., 1967) and basic affect (+5 very good to –5 very bad) (Williams et al., 2008) were recorded before and after the first and final exercise plus immersion sessions for each intervention. Participants also completed an 8-item Physical Activity Enjoyment Scale after the first and final sessions, which included subdomains of pleasure, fun, pleasant, invigorating, gratifying, exhilarating, stimulating and refreshing (seven positive to one negative) (Mullen et al., 2011). These data were used to characterise basic psychological and physiological responses to the acute heating stimulus and can be found in the Supporting information (Table S2).

## Statistical analysis

Data normality for the dependent variable and covariate was confirmed through skewness and kurtosis prior to analysis. Outliers were removed from analysis if they exceeded ±3 SD from the group mean (Miller, 1991) (one IL-6 data point in the EX+HWI group). In addition, one participant was removed from the brachial artery flow-mediated dilatation analyses in the EX+TWI group as a result of technical issues with the ultrasound. Non-normally distributed data were transformed using a $log_{10}$ transformation to reduce the impact of outliers. If data failed to conform to normality after transformation, non-parametric tests were used ($\dot{V}_{O_2peak}$, peak power, time to exhaustion, IL-10, IL-1$\beta$, ET-1 and VEGF). Parametric data were analysed using a one-way analysis of covariance (ANCOVA), with delta change from pre- to post-intervention as the dependent variable, group as the fixed factor and respective baseline data as the covariate. The ANCOVA approach was selected to adjust for potential baseline differences and because it can provide greater statistical power than ANOVA by accounting for variance explained by the covariate (Van Breukelen, 2006). Further assumptions of ANCOVA were assessed and met prior to analysis, including (1)

no significant effect between the independent variable and covariate and (2) homogeneity of regression between the dependent variable and covariates. Non-parametric data were analysed using Quade's ANCOVA. To do this, pre- to post-intervention delta change was used as the dependent variable and baseline data as the covariate, with both ranked accordingly. A linear regression of the ranks of the dependent variable on the ranks of covariate was analysed to provide unstandardised residuals. A one-way ANCOVA was then used, with the unstandardised residuals as the dependent variable and group as the fixed factor. All FMD% analyses were performed on adjusted values that were allometrically scaled for arterial diameter given that inter-individual differences in arterial diameter can affect subsequent vasoactive responses to occlusion (Atkinson et al., 2013). To perform the adjusted analysis, the change in logarithmically transformed diameter in response to occlusion were analysed using a generalised estimation equation, incorporating baseline diameter as a covariate, with the resulting mean difference transformed to the original units of percentage difference (%). This analysis process was performed in accordance with the established guidelines (Atkinson & Batterham, 2013). To assist with data interpretation, mean and 95% confidence intervals (CI) were calculated for parametric data, whereas the median and quartiles 1 (25th percentile) and 3 (75th percentile) (Q1, Q3) were provided for non-parametric data. In addition, effect sizes were calculated as partial eta squared ($\eta^2_p$) and Hedges' *g* to assist with assessing the practical significance of findings. This approach also enabled us to establish the magnitude of any changes in response to each individual intervention separately, providing clinically relevant insights. For exploratory within-group analyses, *P* values have been omitted to avoid issues with multiple comparisons. The alpha value was a priori set at *P* < 0.05 for all statistical tests. All analyses were completed in SPSS, version 25.0 (IBM Corp., Armonk, NY, USA), Prism, version 9 (GraphPad Software Inc., San Diego, CA, USA) or Excel (Microsoft Corp.).

## Results

### CONSORT 2010 flow diagram

CONSORT 2010 flow diagram (Figure 2), followed by baseline characteristics (Table 1).

### Adverse events

No adverse events were documented in the current study. However, three participants in the EX+HWI group did suffer from reoccurring mild presyncopal symptoms

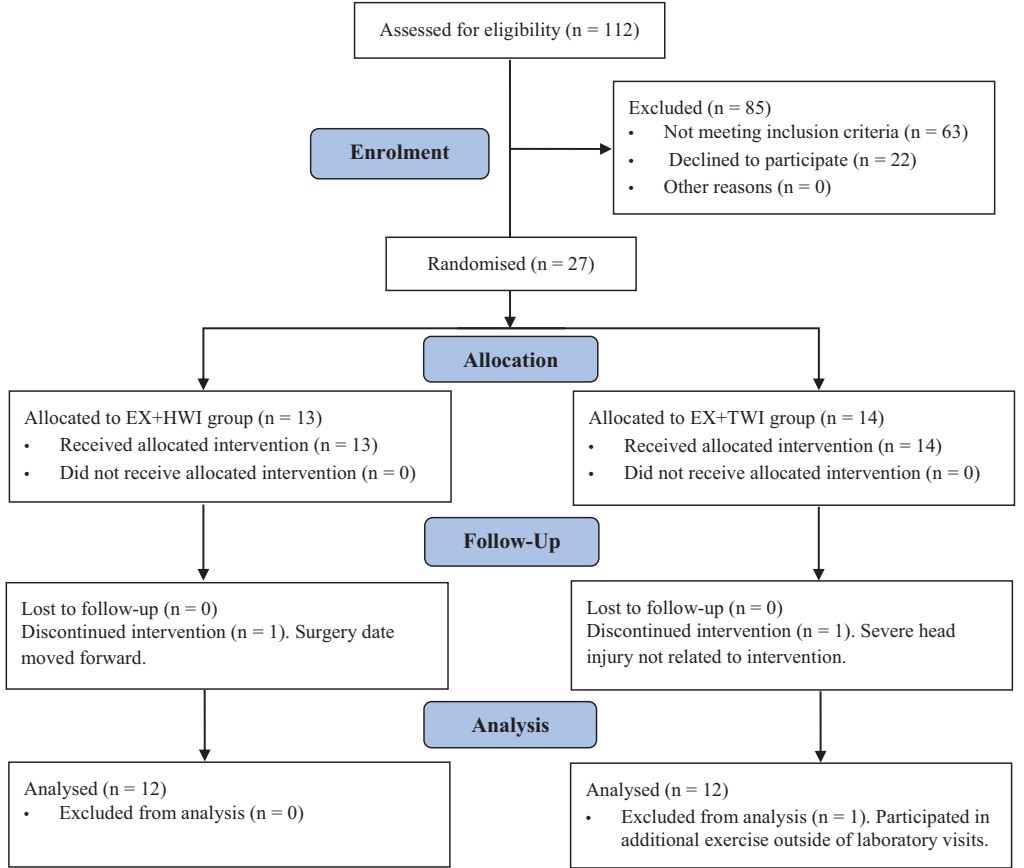

**Figure 2. Consort flow diagram representing the enrolment, allocation, follow-up and analysis of this trial in accordance with the Consolidation Standards of Reporting Trials guidelines (Schulz et al., 2010)**
The CONSORT flow diagram outlines the progression of participants through each stage of the study, including enrolment, random allocation to either the EX+HWI or EX+TWI group, adherence to the 8-week intervention, and completion of pre- and post-intervention assessments. [Colour figure can be viewed at wileyonlinelibrary.com]

**Table 1. Baseline characteristics of the post-exercise hot water immersion (EX+HWI) and post-exercise thermoneutral water immersion (EX+TWI) groups.**

| Characteristic | EX+HWI | EX+TWI |
|---|---|---|
| **Anthropometrics** | | |
| Sex (females) | 12 (6) | 12 (7) |
| Age (years) | 59 ± 3 | 57 ± 6 |
| Weight (kg) | 79 ± 15 | 77 ± 9 |
| Height (cm) | 169 ± 7 | 165 ± 8 |
| BMI (kg m$^{-2}$) | 27 ± 3 | 28 ± 3 |
| Systolic BP (mmHg) | 132 ± 13 | 126 ± 14 |
| Diastolic BP (mmHg) | 84 ± 9 | 82 ± 9 |
| $\dot{V}_{O_2 peak}$ (mL kg$^{-1}$ min$^{-1}$) | 19 ± 3 | 20 ± 5 |
| **Antihypertensive medications** | | |
| Ramipril (*n*) | 1 | 1 |
| Losartan (*n*) | 1 | 0 |
| Bisoprolol (*n*) | 0 | 1 |
| **Lipid lowering medications** | | |
| Simvastatin (*n*) | 1 | 0 |
| Atorvastatin (*n*) | 0 | 1 |

Anthropometrics, antihypertensives, and lipid lowering medications in the EX+HWI (*n* = 12; six females) and EX+TWI (*n* = 12; seven females) groups. Data are presented as the mean ± SD.

throughout the intervention when exiting the hot tub in the form of dizziness and nausea.

## Adherence

In the EX+HWI group, four sessions were missed resulting in an adherence rate of 99% and six sessions were missed in the EX+TWI group, which equated to an adherence of 98%. Two participants withdrew from the study because of unrelated external factors (injury and surgery), resulting in a total of 24 participants taking part in the present study.

## Anthropometric measures

There were no statistical differences between-groups for changes in nude mass [0.74 (–0.57–2.05) kg, $P = 0.253$, $\eta^2_p = 0.062$] or body mass index [0.33 (–0.16 to 0.81) kg m$^{-2}$, $P = 0.163$, $\eta^2_p = 0.090$]. Within-group results showed no changes in nude mass [–0.53 (–1.36 to 0.30) kg, $g = -0.383$] or body mass index [–0.22 (–0.51 to 0.07) kg m$^{-2}$, $g = -0.460$] in the EX+HWI group. Likewise, there were no changes in nude mass [0.22 (–0.61 to 1.37) kg, $g = 0.225$] or body mass index [0.09 (–0.24 to 0.51) kg m$^{-2}$, $g = 0.214$] in the EX+TWI group.

## Cardiorespiratory fitness

There were no significant differences between-groups for changes in $\dot{V}_{O_2\text{peak}}$ [0.9 (Q1: –1.3, Q3: 1.6) mL kg$^{-1}$ min$^{-1}$, $P = 0.623$, $\eta^2_p = 0.012$], with the EX+HWI group showing a mean increase of 2.7 (1.7–3.5) mL kg$^{-1}$ min$^{-1}$ ($g = 1.676$) compared to 2.5 (1.4–3.2) mL kg$^{-1}$ min$^{-1}$ ($g = 1.507$) in the EX+TWI group. Similarly, no significant between-group differences were observed in VT$_2$ [0.7 (–2.3 to 3.7) mL kg$^{-1}$ min$^{-1}$, $P = 0.639$, $\eta^2_p = 0.011$] and VT$_1$ [0.7 (–1.4 to 2.9) mL kg$^{-1}$ min$^{-1}$, $P = 0.496$, $\eta^2_p = 0.022$]. Within-group results indicated no change in VT$_2$ in the EX+HWI [1.7 [–0.7 to 4.0) mL kg$^{-1}$ min$^{-1}$, $g = 0.417$] and EX+TWI [1.2 (–1.7 to 4.0) mL kg$^{-1}$ min$^{-1}$, $g = 0.249$] groups. By contrast, there was a mean increase in VT$_1$ for both the EX+HWI [2.1 (0.1–4.0) mL kg$^{-1}$ min$^{-1}$, $g = 0.669$] and EX+TWI [1.5 (0.1–2.8) mL kg$^{-1}$ min$^{-1}$, $g = 0.633$] groups. Finally, there were no significant between-group differences in peak power [1 (Q1: –4, Q3: 5) W, $P = 0.976$, $\eta^2_p < 0.001$] and time to exhaustion [0 (Q1: –19, Q3: 48) s, $P = 0.737$, $\eta^2_p = 0.005$]. Within-group results showed a mean increase in peak power [16 (11–21) W, $g = 1.636$] and time to exhaustion [63 (38–80) s, $g = 1.676$] in the EX+HWI group. Similarly, an increase in peak power [16 (10–22) W, $g = 1.682$] and time to exhaustion [62 (44–95) s, $g = 1.607$] was also observed in the EX+TWI group.

## Blood pressure

The EX+HWI group had a significantly larger change in MAP [–4 (–8 to 0) mmHg, $P = 0.029$, $\eta^2_p = 0.207$] (Fig. 3*C*) and DBP [–4 (–8 to –1) mmHg, $P = 0.027$, $\eta^2_p = 0.211$] (Fig. 3*B*) compared to the EX+TWI group. By contrast, there were no significant differences between-groups for changes in SBP [–4 (–11 to 2) mmHg, $P = 0.144$, $\eta^2_p = 0.099$] (Fig. 3*A*) or PP [0 (–5 to 5) mmHg, $P = 0.999$, $\eta^2_p < 0.001$] (Fig. 3*D*). Within-group results showed a mean decrease in MAP [–6 (–8 to –3) mmHg, $g = -1.405$], SBP [–8 (–12 to –4) mmHg, $g = -1.259$] and DBP [–5 (–8 to –2) mmHg, $g = -1.077$], but not PP [–3 (–7 to 1) mmHg, $g = -0.479$] in the EX+HWI group. By contrast, there were no changes in MAP [–2 (–5 to 1) mmHg, $g = -0.386$], SBP [–3 (–8 to 2) mmHg, $g = -0.374$], DBP [–1 (–4 to 1) mmHg, $g = -0.307$] or PP [–2 (–6 to 2) mmHg, $g = -0.237$] in the EX+TWI group.

## Endothelial function

Between-group testing showed a significantly larger change in FMD% in the EX+HWI group compared to the EX+TWI group [2.33% (0.31–5.23%), $P = 0.030$, $\eta^2_p = 0.206$] (Fig. 4*A*), with the EX+HWI group showing a mean increase of 2.77% (0.33–5.21%) ($g = 0.676$) compared to 0.45% (–3.38% to 2.49%) ($g = 0.095$) in the EX+TWI group. There were no significant between-group differences in baseline diameter [0.11 (–0.13 to 0.35) mm, $P = 0.350$, $\eta^2_p = 0.044$], peak diameter [0.22 (–0.03 to 0.47) mm, $P = 0.078$, $\eta^2_p = 0.147$], reactive hyperaemia shear rate area under the curve [–1.12 (–14.95 to 12.74) s$^{-1}$ × 10$^3$, $P = 0.869$, $\eta^2_p = 0.001$] or time to peak FMD [3 (–18 to 13) s, $P = 0.701$, $\eta^2_p = 0.008$]. Data on between- and within-group differences for measures of endothelial function are provided in Table 2.

## Arterial stiffness

There was no significant between-group difference in the change in aortic pulse wave velocity [0.10 (–0.53 to 0.33) m s$^{-1}$, $P = 0.627$, $\eta^2_p = 0.011$] (Fig. 4*B*), with the EX+HWI group showing a mean decrease of –0.29 (–0.51 to –0.06) m s$^{-1}$ ($g = -0.745$) compared to –0.14 (–0.54 to 0.26) m s$^{-1}$ ($g = -0.213$) in the EX+TWI group. There were also no between-group differences for changes in brachial artery beta stiffness index [2.39 (–8.53 to 3.75) a.u., $P = 0.427$, $\eta^2_p = 0.030$], dynamic compliance [0.001 (–0.001 to 0.004) mm$^2$ mmHg, $P = 0.318$, $\eta^2_p = 0.047$] or conductance [0.27 (–0.97 to 0.43) mL min$^{-1}$ mmHg$^{-1}$, $P = 0.430$, $\eta^2_p = 0.031$]. Data on between- and within-group differences for measures of arterial stiffness are provided in Table 2.

## Circulating glucose, cholesterol, inflammatory, angiogenic, and vasoactive markers

There were no significant differences between-groups for changes in circulating glucose (fasted and tolerance), cholesterol (low-density lipoproteins, high-density lipoproteins, triglycerides and total cholesterol), inflammatory (CRP, TNF-$\alpha$, IL-6, IL-10 and IL-1$\beta$) and vasoactive markers ET-1, VEGF and MMP-9). Data on between- and within-group differences for circulating blood markers are provided in Table 3.

## Framingham risk score

There was no significant between-group difference in the Framingham risk score [−1.48% (−3.27% to 0.31%), $P = 0.099$, $\eta^2_p = 0.124$], with the EX+HWI group showing a mean decrease of −1.92% (−3.25% to −0.59%)

($g = -0.858$) compared to −0.38% (−1.71% to 0.96%) ($g = -0.167$) in the EX+TWI group.

## Quality of life

Between-group testing showed a significantly larger change in the physical health component score in the EX+HWI group compared to the EX+TWI group [5 (Q1: 0, Q3: 15) a.u., $P = 0.036$, $\eta^2_p = 0.211$], but no significant difference between groups in the mental health component score [4 (Q1: 0, Q3: 6) a.u., $P = 0.056$, $\eta^2_p = 0.179$]. Within-group results showed a median increase in the physical health component score [7 (Q1: 2, Q3: 17) a.u., $g = 0.707$] and mental health component score [10 (Q1: 5, Q3: 15) a.u., $g = 0.688$] in the EX+HWI group. By contrast, there was no change in the physical health component score [1 (Q1: 0, Q3: 3) a.u., $g = 0.407$], but there was a median increase in the mental health

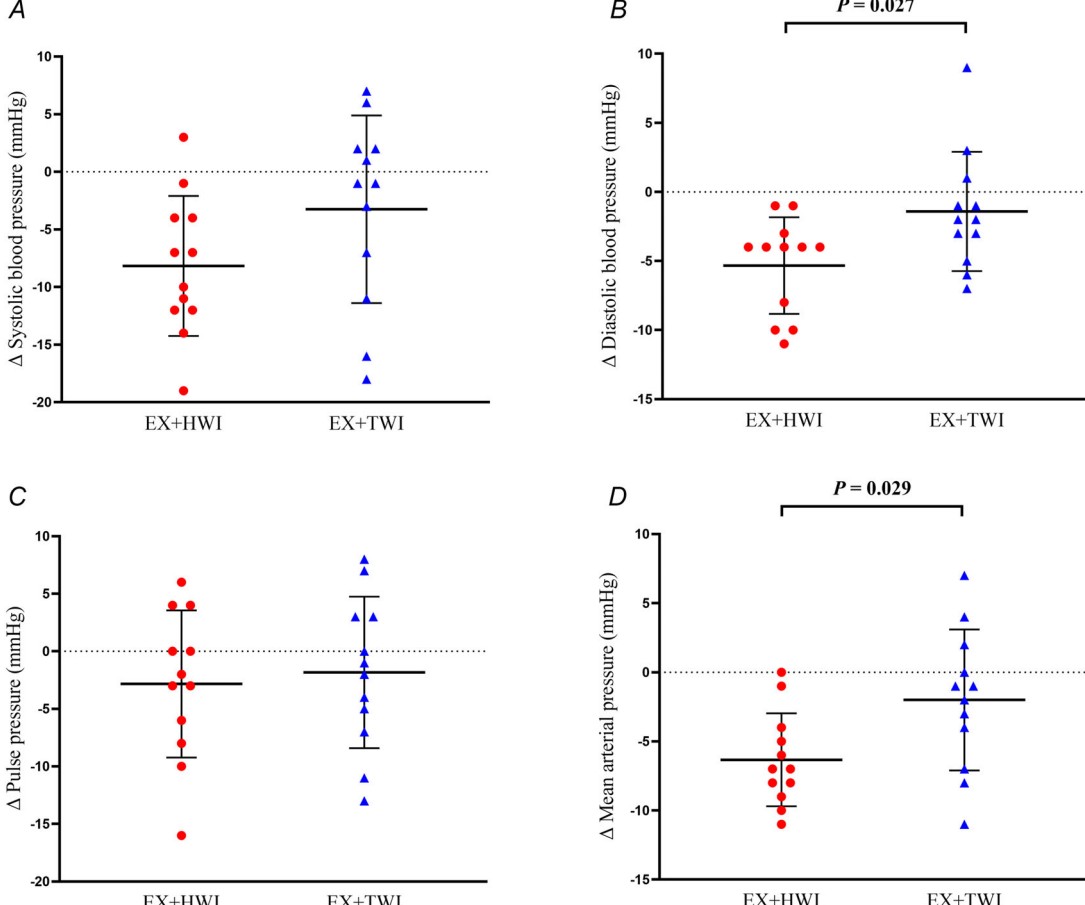

**Figure 3. Changes in systolic blood pressure (*A*), diastolic blood pressure (*B*), pulse pressure (*C*) and mean arterial pressure (*D*) following 8 weeks of EX+HWI (*n* = 12; six females) and EX+TWI (*n* = 12; seven females)**
Blood pressure was measured after 30 min of supine rest and measured in duplicate. Data are presented as the mean ± SD. *P* values represent between-group differences using ANCOVA on delta change scores from pre- to post-intervention, with group as the fixed factor and baseline data as the covariate. [Colour figure can be viewed at wileyonlinelibrary.com]

component score [5 (Q1: 1, Q3: 7) a.u., $g = 0.455$] in the EX+TWI group. Data on between- and within-group differences for quality of life domains are provided in the Supporting information (Table S1).

## Discussion

This is the first study to investigate whether the use of hot water immersion after exercise can augment exercise-derived improvements in cardiovascular and metabolic health. The current findings partially support our hypotheses by showing that post-exercise hot water immersion, compared to exercise followed by thermoneutral water immersion: (1) led to a greater reduction in diastolic and mean arterial blood pressure; (2) increased brachial artery endothelial function; and (3) resulted in a greater improvement in perceived physical health. However, there were no additional improvements in cardiorespiratory fitness, arterial stiffness, circulating glucose, lipids or markers of chronic low-grade inflammation following post-exercise hot water immersion.

### Blood pressure

Post-exercise hot water immersion resulted in a greater reduction in MAP and DBP by 4 mmHg compared to post-exercise thermoneutral water immersion. Although EX+HWI resulted in an 8 mmHg reduction in SBP, this change was not significantly different from the EX+TWI group. However, we observed meaningful reductions in the majority of blood pressure parameters within the post-exercise hot water immersion group, which were not apparent following post-exercise thermoneutral water immersion. Collectively, these findings are clinically relevant because a 2 mmHg reduction in DBP reportedly lowers the risk of coronary heart disease and stroke by 6% and 15%, respectively (Cook, 1995). In addition, previous work has demonstrated that a 5 mmHg reduction in SBP reduces the risk of future CVD events by 10% (Rahimi et al., 2021). According to the American Heart Association, these blood pressure reductions resulted in three EX+HWI participants moving from Stage 2 (>140 or >90 mmHg) to Stage 1 (130–139 or 80–89 mmHg) Hypertensive, one from Stage 1 to Elevated (120–129 and <80 mmHg) and one from Elevated to Normal (<120 and 80 mmHg). In the EX+TWI group, one participant went from Stage 1 to Normal and one from Elevated to Normal. These improvements in blood pressure indicate that post-exercise hot water immersion may serve as a practical strategy for managing hypertension or preventing its onset in at-risk individuals.

The synergistic haemodynamic effects observed in the present study are broadly consistent with previous studies showing that 8–12 weeks of heat therapy combined with exercise leads to greater blood pressure reductions compared to exercise alone (Akerman et al., 2019; Lee et al., 2022). However, one notable difference is that both previous studies reported a larger reduction in SBP alone, whereas we observed a significantly greater reduction in DBP and MAP following post-exercise heating, with no further decrease in SBP. Our observations align with those of Brunt et al. (2016), who reported

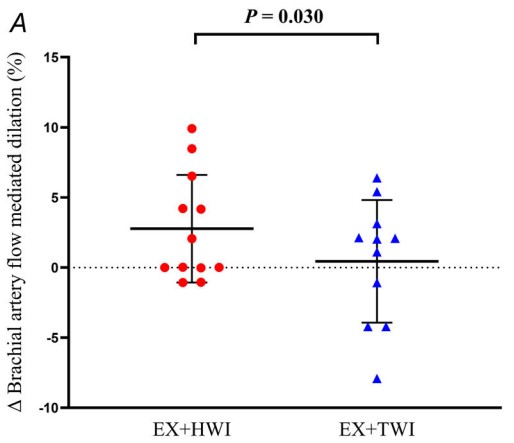
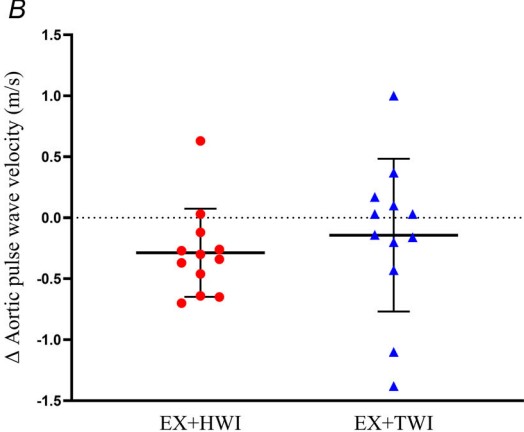

**Figure 4. Changes in brachial artery flow-mediated dilatation (*A*) and aortic pulse wave velocity (*B*) following 8 weeks of EX+HWI (*n* = 12; six females) and EX+TWI (*n* = 11; six females)**
Brachial artery flow-mediated dilation and aortic pulse wave velocity were measured after 10 min of supine rest, with the latter measured in duplicate. Data are presented as the mean ± SD. *P* values represent between-group differences using ANCOVA on delta change scores from pre- to post-intervention, with group as the fixed factor and baseline data as the covariate.1 [Colour figure can be viewed at wileyonlinelibrary.com]

**Table 2. Changes in brachial artery flow-mediated dilatation and stiffness parameters following 8 weeks of post-exercise hot water immersion (EX+HWI) and post-exercise thermoneutral water immersion (EX+TWI).**

| | EX+HWI | | | EX+TWI | | | Statistical significance |
| --- | --- | --- | --- | --- | --- | --- | --- |
| | Week 0 | Week 8 | Pre to post mean difference and 95% CIs | Week 0 | Week 8 | Pre to post mean difference and 95% CIs | Delta change between-groups |
| **Brachial artery flow-mediated dilatation** | | | | | | | |
| Baseline diameter (mm) | 3.97 ± 0.81 | 4.11 ± 0.89 | 0.14 (−0.04 to 0.33) | 4.11 ± 0.87 | 4.14 ± 0.87 | 0.03 (−0.12 to 0.18) | $P = 0.350$ |
| Peak diameter (mm) | 4.13 ± 0.85 | 4.36 ± 0.93 | 0.23 (0.04–0.41) | 4.37 ± 0.91 | 4.38 ± 0.87 | 0.01 (−0.17 to 0.18) | $P = 0.078$ |
| Absolute FMD (mm) | 0.16 ± 0.07 | 0.25 ± 0.17 | 0.09 (0–0.24) | 0.26 ± 0.11 | 0.24 ± 0.12 | −0.02 (−0.10 to 0.09) | $P = 0.527$ |
| Reactive Hyperaemia SRAUC ($s^{-1} \times 10^3$) | 18 ± 12 | 18 ± 14 | 0.07 (−7.36 to 7.50) | 24 ± 17 | 22 ± 19 | −2.07 (−11.69 to 7.54) | $P = 0.869$ |
| Time to peak FMD (s) | 38 ± 20 | 43 ± 17 | 5 (−9 to 17) | 37 ± 15 | 45 ± 18 | 8 (−8 to 24) | $P = 0.701$ |
| Beta stiffness index (a.u.) | 19.86 ± 5.71 | 19.14 ± 5.28 | −0.72 (−3.53 to 2.09) | 21.03 ± 9.13 | 22.36 ± 10.97 | 1.32 (−4.74 to 7.38) | $P = 0.427$ |
| Dynamic compliance ($mm^2$ mmHg) | 0.013 ± 0.007 | 0.015 ± 0.007 | 0.001 (0.001–0.002) | 0.012 ± 0.003 | 0.012 ± 0.004 | −0.001 (−0.003 to 0.002) | $P = 0.318$ |
| Conductance (mL $min^{-1}$ $mmHg^{-1}$) | 0.41 ± 0.28 | 0.46 ± 0.25 | 0.05 (−0.18 to 0.29) | 0.58 ± 0.43 | 0.47 ± 0.16 | −0.11 (−0.35 to 0.13) | $P = 0.430$ |

Brachial artery flow-mediated dilatation and stiffness responses in the EX+HWI ($n = 12$; six females) and EX+TWI ($n = 12$; seven females) groups. Data in EX+TWI group were analysed on ($n = 11$; six females) for measures of absolute FMD, peak diameter, reactive hyperaemia SRAUC and time to peak FMD. Abbreviations include brachial artery flow-mediated dilatation FMD and shear rate area under the curve (SRAUC). Data are expressed as the mean ± SD. *P* values represent between-group differences using ANCOVA on delta change scores from pre- to post-intervention, with group as the fixed factor and baseline data as the covariate.

that repeated hot water immersion reduces DBP and MAP, but not SBP in normotensive young adults. The cause of the inconsistency between studies could relate to differences in heating modalities (sauna *vs.* hot water immersion) and/or pre-intervention resting blood pressure levels, which have been suggested as potential factors that influence the antihypertensive effects of heat therapy (Pizzey et al., 2021). Regardless, the clinically relevant reductions in blood pressure observed in the current study demonstrate the antihypertensive potential of combining exercise training and hot water immersion. Although exercise remains a mainstay first-line strategy to prevent hypertension, the typically low adherence to long-term regimes, and the lack of additional blood pressure reduction from high-intensity compared to moderate-intensity exercise (Costa et al., 2018), highlight the value of adjunct non-pharmacological strategies, such as hot water immersion, to optimise the antihypertensive effects of exercise.

### Endothelial function and arterial stiffness

To our knowledge, this is the first study to demonstrate that post-exercise heating elicits a greater improvement in brachial artery endothelial function than post-exercise thermoneutral water immersion. We showed that 8 weeks of EX+HWI improved FMD by 2.3% more than EX+TWI, which reportedly confers an 18–30% reduced risk of future CVD events (Thijssen et al., 2019). Given that endothelial dysfunction is an early and independent predictor of CVD risk (Thijssen et al., 2019), the present study clearly shows the synergistic potential of combining hot water immersion and exercise training as an effective strategy to optimise overall cardio-protection during middle age. Our findings differ from earlier research, where 12 weeks of 39°C whole-body immersion prior to calisthenics exercise (Akerman et al., 2019) and 8 weeks of 42°C lower limb immersion following moderate-intensity exercise (Cheng et al., 2025) resulted

**Table 3. Changes in circulating glucose, lipids, inflammatory, angiogenic, and vasoactive markers following 8 weeks of post-exercise hot water immersion (EX+HWI) and post-exercise thermoneutral water immersion (EX+TWI).**

| | EX+HWI | | | EX+TWI | | | Statistical significance |
|---|---|---|---|---|---|---|---|
| | Week 0 | Week 8 | Pre to post mean/median difference and 95% CIs/Q1–Q3 | Week 0 | Week 8 | Pre to post mean/median difference and 95% CIs/Q1–Q3 | Delta change between-groups |
| **Glucose and lipid markers** | | | | | | | |
| Fasted glucose (mmol L⁻¹) | 4.71 ± 0.49 | 4.71 ± 0.31 | 0 (−0.36 to 0.37) | 4.61 ± 0.40 | 4.55 ± 0.39 | −0.06 (−0.35 to 0.13) | $P = 0.638$ |
| Glucose challenge (AUC) | 254 ± 48 | 250 ± 36 | −4 (−22 to 12) | 247 ± 35 | 248 ± 37 | 1 (−18 to 20) | $P = 0.745$ |
| Low-density lipoproteins (mmol L⁻¹) | 2.65 ± 1.12 | 2.55 ± 1.41 | −0.10 (−0.92 to 0.72) | 3.66 ± 1.52 | 3.56 ± 1.31 | −0.10 (−1.04 to 0.84) | $P = 0.393$ |
| High-density lipoproteins (mmol L⁻¹) | 1.03 ± 0.39 | 0.93 ± 0.25 | −0.10 (−0.39 to 0.19) | 1.03 ± 0.51 | 0.96 ± 0.44 | −0.06 (−0.39 to 0.26) | $P = 0.799$ |
| Triglycerides (mmol L⁻¹) | 0.69 ± 0.29 | 0.75 ± 0.34 | 0.06 (−0.13 to 0.25) | 1.06 ± 0.52 | 0.99 ± 0.52 | −0.07 (−0.49 to 0.35) | $P = 0.591$ |
| Total cholesterol (mmol L⁻¹) | 3.82 ± 1.12 | 3.55 ± 1.22 | −0.26 (−1.33 to 0.80) | 4.60 ± 1.73 | 4.44 ± 1.54 | −0.16 (−1.37 to 1.04) | $P = 0.270$ |
| **Inflammatory markers** | | | | | | | |
| C-reactive protein (mg L⁻¹) | 1.35 ± 0.69 | 1.37 ± 0.92 | 0.02 (−0.60 to 0.62) | 1.12 ± 0.68 | 1.34 ± 0.70 | 0.22 (−0.14 to 0.60) | $P = 0.704$ |
| Tumour necrosis factor-α (pg mL⁻¹) | 6.30 ± 1.23 | 6.47 ± 1.19 | 0.17 (−0.16 to 0.48) | 7.18 ± 3.59 | 7.19 ± 3.31 | 0.01 (−1.03 to 1.04) | $P = 0.977$ |
| Interleukin-6 (pg mL⁻¹) | 0.67 ± 0.43 | 0.65 ± 0.28 | −0.02 (−0.19 to 0.10) | 0.77 ± 0.60 | 0.75 ± 0.59 | −0.02 (−0.38 to 0.33) | $P = 0.733$ |
| Interleukin-10 (pg mL⁻¹) | 0.29 ± 0.16 | 0.27 ± 0.09 | −0.02 (Q1: −0.04, Q3: 0.04) | 0.31 ± 0.18 | 0.31 ± 0.20 | 0 (Q1: −0.01, Q3: 0.03) | $P = 0.612$ |
| Interleukin-1β (pg mL⁻¹) | 0.33 ± 0.25 | 0.31 ± 0.36 | −0.02 (Q1: −0.02, Q3: 0.14) | 0.30 ± 0.06 | 0.50 ± 0.59 | 0.20 (Q1: 0, Q3: 0.53) | $P = 0.139$ |
| **Angiogenic and vasoactive markers** | | | | | | | |
| Endothelin-1 (pg mL⁻¹l) | 1.24 ± 0.52 | 1.42 ± 0.27 | 0.18 (Q1: 0.11, Q3: 0.25) | 1.60 ± 0.43 | 1.33 ± 0.76 | −0.27 (Q1: −0.28, Q3: 0.15) | $P = 0.640$ |
| Vascular endothelial growth factor (pg mL⁻¹) | 46 ± 23 | 43 ± 24 | −3 (Q1: −6, Q3: 0) | 43 ± 63 | 36 ± 72 | −7 (Q1: −7, Q3: 2) | $P = 0.980$ |
| Cortisol (ng mL⁻¹) | 109 ± 33 | 128 ± 47 | 19 (−6 to 46) | 106 ± 29 | 115 ± 36 | 9 (−5 to 23) | $P = 0.431$ |
| Matrix metalloproteinase-9 (ng mL⁻¹) | 229 ± 126 | 221 ± 85 | −8 (−79 to 64) | 241 ± 107 | 187 ± 61 | −54 (−105 to −3) | $P = 0.146$ |

Concentrations of circulating serum markers in the EX+HWI ($n = 12$; six females) and EX+TWI ($n = 12$; seven females) groups. Data are expressed as the mean ± SD or medians ± IQR. $P$ values represent between-group differences using ANCOVA or Quade's ANCOVA on delta change scores from pre- to post-intervention, with group as the fixed factor and baseline data as the covariate.

in no change in brachial artery FMD in patients with peripheral arterial disease and young healthy adults, respectively. The reasons for these discrepancies remain unclear, but they probably relate to inherent differences in baseline endothelial (dys)function across the respective study populations, as well as the use of less intense combinations of heating and exercise stimuli compared to the present study. Indeed, episodic increases in shear stress are the primary stimulus for endothelial adaptation in response to both exercise training and passive heating (Carter et al., 2014; Green et al., 2017), which is probably more pronounced during more intensive exercise and heating bouts.

By contrast to endothelial function, no differences were observed in the changes in aortic or brachial artery stiffness parameters between the EX+HWI and EX+TWI groups. These findings correspond with previous studies reporting no additional benefit of using heat therapy as an adjunct to exercise on aortic stiffness (Akerman et al., 2019; Cheng et al., 2025; Lee et al., 2022). As is the case with isolated exercise-induced arterial remodelling (Green et al., 2017), these data indicate that functional adaptation may occur prior to any structural alterations in response to post-exercise heating. Similarly, peripheral muscular arteries may be more susceptible, or respond more rapidly to, post-exercise heating than central vessels, which is largely in agreement with the effects of exercise-induced arterial remodelling (Rakobowchuk et al., 2008). However, it is conceivable that a larger post-exercise heating stimulus and/or a longer inter-vention duration could have elicited concomitant peri-pheral and central arterial adaptation. For example, Brunt et al. (2016) demonstrated that a more intense heating protocol consisting of four or five immersion sessions per week, each lasting 90 min at 40.5°C and raising core body temperature by ∼1.5°C for 60 min, induced systemic arterial remodelling, including improvements in brachial artery endothelial function, femoral artery compliance, aortic stiffness and carotid intima-media thickness. Regardless, the present study demonstrates that a short-term moderate-intensity exercise training regime, which aligns with the minimum weekly physical activity recommendations, offers negligible vascular benefit in physically inactive middle-aged adults. However, the addition of 30 min of hot-water immersion following exercise appears synergistic and provides augmented vascular effects. Furthermore, this additive endothelial benefit was accompanied by further reductions in blood pressure, despite no further improvements in cardiorespiratory fitness or reductions in weight, circulating glucose, lipids or inflammatory markers. Because the vascular endothelium plays a crucial role in regulating vascular tone and maintaining optimal blood pressure (AlGhatrif & Lakatta, 2015; Green et al., 2017), post-exercise hot water immersion appears to offer direct therapeutic benefits for vascular haemodynamics, independent of other traditional CVD risk factors.

## Cardiorespiratory fitness

Both groups in this study showed an increase in $\dot{V}_{O_2peak}$ (EX+HWI 2.7 mL kg$^{-1}$ min$^{-1}$ vs. EX+TWI: 2.5 mL kg$^{-1}$ min$^{-1}$), but there were no additional improvements between the EX+HWI and EX+TWI groups. The increase in $\dot{V}_{O_2peak}$ was comparable in magnitude (∼2 mL kg$^{-1}$ min$^{-1}$) to the improvement observed by Bailey et al. (2016), who observed similar increases in cardiorespiratory fitness following either regular exercise or hot water immersion over 8 weeks. Indeed, we had hypothesised that combining the exercise and heating stimuli would have an additive effect, leading to a greater improvement in cardio-respiratory fitness in the EX+HWI group. Our hypothesis was based on prior research suggesting heating can stimulate mechanisms that underpin exercise-induced improvements in cardiorespiratory fitness, including skeletal muscle capillarisation (Hesketh et al., 2019), plasma volume expansion (Scoon et al., 2007) and increases in haemoglobin mass (Lundby et al., 2023). The present findings are consistent with recent work by Cheng et al. (2025), who also demonstrated no added benefit from 8 weeks of post-exercise heating, although their study used a lower limb heating stimulus, which may have a lesser physiological effect than the water immersion protocol used in our study. By contrast, studies using post-exercise sauna (Kirby et al., 2021; Lee et al., 2022) have reported greater improvements in fitness, although it is unclear why sauna might be more effective than hot water immersion. It is possible that the heating intensity or duration in our study was insufficient to induce the necessary adaptations. Further studies focusing on the mode, intensity, and duration of combined exercise and heating stimuli are needed to clarify these effects and may require a more focussed study of the underpinning mechanisms.

## Inflammation and metabolic health

Taking part in 30 min of moderate-intensity aerobic exercise, on average three times per week, followed by 30 min of hot water immersion (40°C), did not improve circulating markers of cardiometabolic health over a period of 8 weeks compared to post-exercise thermo-neutral water immersion (34°C). Although there are limited longitudinal data with which to compare our findings, our results contrast with those by Ely, Clayton, et al. (2019) who reported lowered serum IL-6, TNF-α, fasting glucose and cholesterol following 8–10 weeks of repeated hot water immersion. Indeed, this was surprising

given that previous work from our laboratory had shown that the post-exercise heating protocol used in the current study did elicit a greater acute anti-inflammatory response than exercise alone (Steward et al., 2024). Acute anti-inflammatory responses are considered to be an important mechanism by which long-term reductions in chronic low-grade inflammation are mediated (Gleeson et al., 2011); however, the magnitude of these responses may have been too small to induce longitudinal adaptations. By comparison, the dose of heating was considerably higher in the study by Ely, Clayton, et al. (2019) who exposed participants to 60 min of water immersion at a temperature of 40.5°C. Indeed, other studies of shorter-term heating interventions also support the hypothesis that a greater heating stimulus may be required for metabolic benefits. For example, Pallubinsky et al. (2020) employed a considerably longer duration of heat exposure than the current study, which comprised of 4–6 h of exposure to 34°C air per day over a period of 10 days, and reported significant improvements in glucose metabolism in overweight adults. It should also be acknowledged that the sample population in the present study had low levels of inflammatory markers at baseline, which would make further improvements less probable. To date, rapid improvements in inflammatory markers, including CRP, TNF-$\alpha$, IL-6 and IL-1$\beta$, have only been observed after heat therapy in clinical populations with polycystic ovary syndrome and chronic heart failure, where the participants presented with markedly higher levels of inflammation at baseline (Ely, Clayton, et al., 2019; Oyama et al., 2013). Taken together, this may suggest diseases characterised by chronic low-grade inflammation and/or insulin resistance, such as rheumatoid arthritis and type 2 diabetes, may benefit more from heating-based or combined exercise and heating-based interventions. However, the duration and intensity of the heating stimulus used may need to be greater than that used in the present study.

## Limitations

The present study has several limitations that should be considered when interpreting our results. First, longer-term studies are required to establish whether the observed changes in cardiovascular health markers translate into a reduction in CVD incidence. Although we observed significant improvements in DBP, MAP, and brachial artery FMD in the EX+HWI compared to the EX+TWI group, the present study was not powered to detect effects in all secondary and exploratory outcome measures. In addition, it remains unclear whether the improvements in brachial artery blood pressure and endothelial function would translate to other healthy or clinical populations. In this regard,

further studies are necessary to establish the ideal dose and mode of post-exercise heat therapy to enhance not only overall health, but also specific cardiovascular and cardiometabolic end-point measures related to the pathophysiology of different chronic diseases.

## Perspectives

Post-exercise heating provided additional improvements in blood pressure and increased brachial artery endothelial function, which also translated into a greater improvement in perceived physical health. The beneficial effects were apparent without changes in other CVD risk factors, which demonstrates that post-exercise heating can elicit clinically relevant cardiovascular benefits even if the stimulus is insufficient to induce additional improvements in cardiorespiratory fitness, metabolic health or inflammation. This is particularly important given the poor adherence to exercise guidelines in this demographic (physically inactive middle-aged adults). Although it is worth highlighting that these were reductions in supine blood pressure and not seated or 24 h blood pressures, which are more commonly used in clinical practice. When considering evidence from other studies, it appears that improvements in blood pressure and endothelial function require a significantly smaller heating stimulus than that required to induce benefits to metabolic health or inflammation, which might in turn impact the safety of the intervention and long-term adherence (Steward et al., 2023). Although adherence (for both groups) was excellent in the present study, some participants in the heating group reported persistent mild negative presyncopal symptoms (i.e. dizziness). As such, future studies should endeavour to study the feasibility, adherence and efficacy of similar post-exercise heating protocols under free-living conditions outside of the laboratory on a larger scale.

## Conclusions

In summary, these findings provide novel evidence that 8 weeks of post-exercise hot water immersion further improves diastolic blood pressure, mean arterial blood pressure and brachial artery endothelial function compared to post-exercise thermoneutral water immersion. These vascular and haemodynamic benefits were apparent in the absence of any additional improvements in other traditional CVD risk factors, such as cardiorespiratory fitness, circulating glucose, lipids and inflammatory markers, in middle-aged adults who do not achieve the minimum recommended physical activity guidelines.

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

## Additional information

### Data availability statement

Data are available from the corresponding author upon reasonable request.

### Competing interests

The authors declare that they have no competing interests.

### Author contributions

C.S, T.C, C.P, D.T and M.H were responsible for the conception and design of the study. C.S, C.M, S.R and T.C were responsible for data acquisition, whereas D.T, C.P and M.H also assisted in interpretation of the data. All authors contributed to drafting or revision of the written work and approved the final version submitted for publication. The authors agree to be accountable for all aspects of the work in ensuring that questions related to the accuracy or integrity of any part of the work are appropriately investigated and resolved. All persons designated as authors qualify for authorship, and all those who qualify for authorship are listed.

### Funding

The authors received no external funding to support the work.

### Keywords

blood pressure, cardiovascular health, inflammation, moderate-intensity exercise, passive heating, physical inactivity, vascular function

### Supporting information

Additional supporting information can be found online in the Supporting Information section at the end of the HTML view of the article. Supporting information files available:

**Peer Review History**
**Table S1**
**Table S2**

