## [Peer Review History · The Journal of Physiology]

Post-exercise hot water immersion enhances vascular benefits of exercise without further improving cardiorespiratory fitness, lipids, or inflammation

Charles James Steward, Mathew Hill, Campbell Menzies, Sophie Lauren Russell, C Doug Thake, Christopher Pugh, and Tom Cullen

DOI: 10.1113/JP288873

Corresponding author(s): Charles Steward (charles.steward@nottingham.ac.uk)

The following individual(s) involved in review of this submission have agreed to reveal their identity: Joseph Costello (Referee #1); Daniel Gagnon (Referee #2); Sean Williams (Referee #3)

Review Timeline:

Submission Date:	14-Mar-2025
Editorial Decision:	10-Apr-2025
Revision Received:	19-May-2025
Editorial Decision:	10-Jun-2025
Revision Received:	16-Jun-2025
Editorial Decision:	23-Jun-2025
Revision Received:	24-Jun-2025
Accepted:	01-Jul-2025

Senior Editor: Karyn Hamilton

Reviewing Editor: Zachary Schlader

Transaction Report:

Dear Dr Steward,

Re: JP-RP-2025-288873 "Post-exercise hot water immersion enhances vascular benefits of exercise without further improving cardiorespiratory fitness, lipids, or inflammation" by Charles James Steward, Mathew Hill, Campbell Menzies, Sophie Lauren Russell, C Doug Thake, Christopher Pugh, and Tom Cullen

Thank you for submitting your manuscript to The Journal of Physiology. It has been assessed by a Reviewing Editor and by 2 expert referees and we are pleased to tell you that it is potentially acceptable for publication following satisfactory major revision.

REVISION CHECKLIST:

We look forward to receiving your revised submission.

Yours sincerely,

Karyn Hamilton
Senior Editor
The Journal of Physiology

REQUIRED ITEMS

- The contact information for the person responsible for 'Research Governance' at your institution needs to be provided (i.e. David Broom's institutional email address, please). Please ensure the contact is not an author on this paper and provide an alternate contact if necessary, or confirm in the submission form that the author whose email was provided has sole responsibility for research governance. This is the person who is responsible for regulations, principles and standards of good practice in research carried out at the institution, for instance the ethical treatment of animals, the keeping of proper experimental records or the reporting of results.
- The Journal of Physiology funds authors of provisionally accepted papers to use the premium BioRender site to create high resolution schematic figures. Follow this link and enter your details and the manuscript number to create and download figures. Upload these as the figure files for your revised submission. If you choose not to take up this offer, we require figures to be of similar quality and resolution. If you are opting out of this service to authors, state this in the Comments section on the Detailed Information page of the submission form. The link provided should only be used for the purposes of this submission. Authors will be charged for figures created on this premium BioRender account if they are not related to this manuscript submission.
- Please upload separate high-quality figure files via the submission form.
- Please ensure that any tables are editable and in Word format, and wherever possible, embedded in the article file itself.
- Papers must comply with the Statistics Policy: https://jp.msubmit.net/cgi-bin/main.plex?form_type=display_requirements#statistics.

In summary:

- If $n \leq 30$, all data points must be plotted in the figure in a way that reveals their range and distribution. A bar graph with data points overlaid, a box and whisker plot or a violin plot (preferably with data points included) are acceptable formats.
- If $n > 30$, then the entire raw dataset must be made available either as supporting information, or hosted on a not-for-profit repository, e.g. FigShare, with access details provided in the manuscript.
- 'n' clearly defined (e.g. x cells from y slices in z animals) in the Methods. Authors should be mindful of pseudoreplication.
- All relevant 'n' values must be clearly stated in the main text, figures and tables.

- The most appropriate summary statistic (e.g. mean or median and standard deviation) must be used. Standard Error of the Mean (SEM) alone is not permitted.

- Exact p values must be stated. Authors must not use 'greater than' or 'less than'. Exact p values must be stated to three significant figures even when 'no statistical significance' is claimed.

- A Data Availability Statement is required for all papers reporting original data. This must be in the Additional Information section of the manuscript itself. It must have the paragraph heading 'Data Availability Statement'. All data supporting the results in the paper must be either: in the paper itself; uploaded as Supporting Information for Online Publication; or archived in an appropriate public repository. The statement needs to describe the availability or the absence of shared data. Authors must include in their statement: a link to the repository they have used, or a statement that it is available as Supporting Information; reference the data in the appropriate sections(s) of their manuscript; and cite the data they have shared in the References section. Whenever possible, the scripts and other artefacts used to generate the analyses presented in the paper should also be publicly archived. If sharing data compromises ethical standards or legal requirements then authors are not expected to share it, but must note this in their statement. For more information, see our Statistics Policy.

EDITOR COMMENTS

Reviewing Editor:

Comments for Authors to ensure the paper complies with the Statistics Policy (Required):
Precise p-values should be given.

Comments to the Author (Required):

The reviewers both indicated that this manuscript is both timely and important, and the study is comprehensive. Overall, the reviewers liked the study design and outcome measures. Primary issues, however, related to the reporting of clinical trials results. I fully agree with this assessment, but these should be addressable (at least in theory).

Please also see 'Required Items' above.

Senior Editor:

Comments for Authors to ensure the paper complies with the Statistics Policy (Required):

Thank you for providing individual data points and precise p-values within the text. It seems like a straight forward task to provide precise p-values in your figures as well. Please do consider this in your revisions. Thank you.

Comments to the Author:

Thank you for submitting your manuscript for consideration by The Journal of Physiology. As part of the peer review process, we recruited two Referees with expertise in this field of study. I believe that you will see that each took great care in providing you with positive feedback and with important considerations for revising your manuscript to make it even stronger and more transparent. At this point, we would like to invite you to respond point-by-point to the Referees' comments, making corresponding revisions to your manuscript. We look forward to seeing your revised manuscript and thank you for your interest in The Journal of Physiology.

REFEREE COMMENTS

Referee #1:

This study investigatess the effect of post-exercise hot water immersion on vascular function, cardiorespiratory fitness, lipids profile and inflammatory status. The findings provide novel mechanistic evidence that post-exercise hot water immersion lowers diastolic blood pressure, mean arterial pressure, and improves endothelial function, without further enhancing cardiorespiratory fitness, lipid profiles, or inflammation in adults not meeting the minimum physical activity guidelines. The current manuscript is a timely, novel and thought-provoking. In general, the manuscript is well written and the authors should be complimented on undertaking this work in an important and emerging area.

Specific comments

L75 (and elsewhere). The reference to a total of 24 sessions is confusing. Was it a min of 2 and a max 4. Also were any weeks missed. Perhaps some additional detail would be useful (if space permits here, if not add it into the methods)

L79-80. Please consider adding effect sizes for your primary outcome (e.g. BP) here and in the results section.

L248. This is a very good rationale, thank you.

L590. The use 'Con' and 'exercise only' throughout is confusing. Please change to EX+Thermoneutral or EX+TWI throughout. Con suggest a passive intervention which was not the case.

L590-592 and Figure 3. Was this a power issue? Please refer to earlier comment re adding effect sizes

L703. 'Thrice weekly' - see earlier comment. The number of sessions per week needs to be clarified. Please specify this within the method.

Tables. Please highlight any significant effects to make it easy for the reader.

Tables and figures. Please add the statistical test that was performed and effect sizes.

Referee #2:

Overview

The objective of this study was to determine if post-exercise hot water immersion improves cardiorespiratory fitness, vascular health and metabolic health to a greater extent than post-exercise thermoneutral water immersion (lines 138-140). Relatively healthy middle-aged adults were recruited and randomized to 8 weeks of aerobic exercise training followed by immersion in 34 or 40°C water. The outcome variables were measured before and after the intervention. The results show a greater pre/post change in diastolic blood pressure, mean arterial pressure, and flow-mediated dilation in the hot water immersion group (relative to the change in the thermoneutral group). There were no differences in the pre/post change between groups for the other variables. The authors conclude that "8-weeks of post-exercise hot water immersion further improves diastolic blood pressure, mean arterial blood pressure and brachial artery endothelial function. These vascular and haemodynamic benefits were apparent in the absence of any additional improvements in other traditional CVD risk factors, such as cardiorespiratory fitness, circulating lipids and inflammatory markers in physically inactive populations that do not achieve the minimum recommended physical activity guidelines." (lines 754-760).

Main comments

1) I congratulate the authors for conducting this challenging, comprehensive study, and original study. However, I have several concerns regarding the reporting of the study that dampen my enthusiasm. Given this was labelled as a clinical trial, I encourage the authors provide a more transparent reporting of the methods and results. Specifically:

1.1. The manuscript would benefit from a more explicit presentation of primary/secondary/exploratory outcomes. The presentation of the methods, the results and discussion should then be aligned with the importance that was given to each variable. I note that the CT registration lists 25 primary outcomes! Was the study powered to detect the stated effect size for each of these variables? How did the analyses consider multiplicity of testing for so many primary outcomes (no correction is stated in the analysis section of the manuscript).

1.2. There are several inconsistencies between what is reported on the CT registration and what is reported in the

manuscript:

- Lines 185-188: was randomization performed before the pre-intervention assessments? Was randomization performed by someone not involved in the study? Was the sequence known by the study staff? The CT registration states triple blinding; please explain how this was ensured, especially for study participants as I don't see how they could be blinded to hot vs thermoneutral water immersion.

- Some primary outcomes (e.g., glycated haemoglobin) and secondary outcomes (e.g., core temperature, thermal comfort/sensation, affect, physical activity enjoyment) listed on the CT registration are not presented. This gives an impression of selective bias in the reporting of outcomes.

- Some protocol deviations may have occurred and are not presented. For example, the CT registration lists venous blood samples for glycaemic variables but the manuscript states capillary measurements. The age range for inclusion is stated as 45-60 but a range of 45-65 is stated on the CT registration. Lastly, the SF-36 questionnaire is not listed on the CT registration but is presented in the manuscript.

1.3. The objective of the study, as stated on the CT registration and in the manuscript, was to determine if post-exercise hot water immersion provides a "greater" effect than post-exercise thermoneutral water immersion. Accordingly, the analyses, results and interpretation/discussion should be restricted to the between-intervention differences in change scores.

2) Lines 195-197: I appreciate the inclusion of the expected effect size, but it would be important to specify the absolute difference in blood pressure that was expected and/or that was considered important before starting the study. It is also unclear why this effect size is only presented for blood pressure given the uncertainty regarding the primary outcomes and it is also unclear if one measure of blood pressure (systolic, diastolic, mean) was considered more important than others before undertaking the study.

3) By convention, the participant flowchart, the participant characteristics at baseline (please include in main manuscript), the adherence, and any adverse events (currently not stated) should be presented at the start of the results section for a clinical trial.

4) Please include a specific limitations section. I think the most important limitation being that it remains unknown how the observed differences translate into any meaningful effect on longer-term cardiovascular risk/health.

Specific comments

5) Lines 132-135: this sentence is not entirely true since a previous study is available (PMID: 35785965). This study is cited in the discussion, but it would be important to present it in the introduction and briefly explain how the current study differs.

6) Lines 329-332: blinding was performed for FMD measurements, but what about the other outcomes?

7) Line 450: I believe journal policy recommends reporting exact p values when values are greater than 0.05?

END OF COMMENTS

EDITOR COMMENTS

Comment to Editor: We would first like to thank you for taking the time to review and handle our manuscript.

We would also like to inform you that we have revised the manuscript title to better reflect the key findings and outcomes of our study. As a result, the revised title now slightly exceeds the journal's character limit. We kindly ask for your guidance on whether this is acceptable, or if we should revert to the original title to comply with the journal's requirements.

Title: Post-exercise hot water immersion enhances haemodynamic and vascular benefits of exercise without further improving cardiorespiratory fitness, glucose, lipids, or inflammation

Reviewing Editor:

Comments for Authors to ensure the paper complies with the Statistics Policy (Required):
Precise p-values should be given.

Exact p values provided throughout manuscript.

Comments to the Author (Required):

The reviewers both indicated that this manuscript is both timely and important, and the study is comprehensive. Overall, the reviewers liked the study design and outcome measures. Primary issues, however, related to the reporting of clinical trials results. I fully agree with this assessment, but these should be addressable (at least in theory).

We thank the reviewers and editors for their positive comments regarding the manuscript. We agree that there are some inconsistencies between what was entered into the clinical trials registration documentation and what we have reported in the manuscript.

The clinical trials registration was not given the appropriate oversight when it was originally completed, and we should have kept it up to date more effectively. We have made every effort to clarify any discrepancies noted by the reviewer.

Please also see 'Required Items' above.

Senior Editor:

Comments for Authors to ensure the paper complies with the Statistics Policy (Required):

Thank you for providing individual data points and precise p-values within the text. It seems like a straight forward task to provide precise p-values in your figures as well. Please do consider this in your revisions. Thank you.

Precise p values have now been provided throughout the manuscript in the figures and text in response from the comment from reviewer 2.

Comments to the Author:

Thank you for submitting your manuscript for consideration by The Journal of Physiology. As part of the peer review process, we recruited two Referees with expertise in this field of study. I believe that you will see that each took great care in providing you with positive feedback and with important considerations for revising your manuscript to make it even stronger and more transparent. At this point, we would like to invite you to respond point-by-point to the Referees' comments, making corresponding revisions to your manuscript. We look forward to seeing your revised manuscript and thank you for your interest in The Journal of Physiology.

REFEREE COMMENTS

Referee #1:

This study investigates the effect of post-exercise hot water immersion on vascular function, cardiorespiratory fitness, lipids profile and inflammatory status. The findings provide novel mechanistic evidence that post-exercise hot water immersion lowers diastolic blood pressure, mean arterial pressure, and improves endothelial function, without further enhancing cardiorespiratory fitness, lipid profiles, or inflammation in adults not meeting the minimum physical activity guidelines. The current manuscript is a timely, novel and thought-provoking. In general, the manuscript is well written and the authors should be complimented on undertaking this work in an important and emerging area.

We thank the reviewer for their positive comments on the manuscript.

Specific comments

L75 (and elsewhere). The reference to a total of 24 session is confusing. Was it a min or 2 and a max 4. Also were any weeks missed. Perhaps some additional detail would be useful (if space permits here, if not add it into the methods)

The aim was for participants to complete 3 sessions per week, but if one session was missed in a week, an additional session was completed in the following week. We have now made this clearer in the intervention characteristics and discussion sections.

L 237-242: Throughout the 8-week supervised intervention, participants reported to the laboratory at Coventry University on average 3 times per week, either individually or in small groups (<4 participants). If a participant was unable to attend a training session, an additional session was completed the following week, therefore laboratory visits ranged from a minimum of 2 to a maximum of 4 sessions per week. This amounted to 24 training visits consisting of aerobic exercise training and water immersion within the required 8-week period.

L 769-772: As such, taking part in 30 minutes of moderate intensity aerobic exercise, on average 3 times per week, followed by 30 minutes of hot water immersion (40°C) compared to thermoneutral water immersion (34°C), did not improve circulating markers of cardiometabolic health over a period of 8 weeks.

L79-80. Please consider adding effect sizes for you primary outcome (e.g. BP) here and in the results section.

We had originally considered mean/median differences and confidence intervals to be more practically useful in the case of widely understood measurements such as blood pressure, but we have added standardised effect sizes (partial eta squared and Hedges g) as per the reviewer recommendations.

L248. This is a very good rationale, thank you.

Thank you.

L590. The use 'Con' and 'exercise only' throughout is confusing. Please change to EX+Thermoneutral or EX+TWI throughout. Con suggest a passive intervention which was not the case.

We agree and have now changed 'CON' to 'EX+TWI' throughout the manuscript.

L590-592 and Figure 3. Was this a power issues? Please refer to earlier comment re adding effects sizes

Based on the effect sizes, we agree that it does look to be a power issue, but we would like to highlight that the decrease in SBP was still of clinical relevance. As detailed in our previous response, we have added standardised effect sizes.

L 665-666: In addition, previous work has demonstrated that a 5 mmHg reduction in SBP reduces the risk of future CVD events by 10% (Rahimi *et al.*, 2021).

L703. 'Thrice weekly' - see earlier comment. The number of sessions per week needs to be clarified. Please specify this within the method.

As described above, the aim was for participants to complete 3 sessions per week, but if one session was missed in a week, an additional session was completed in the following week to ‘catch up’, so that the total number of sessions was the same for all participants.

L 237-242: Throughout the 8-week supervised intervention, participants reported to the laboratory at Coventry University on average 3 times per week, either individually or in small groups (<4 participants). If a participant was unable to attend a training session, an additional session was completed the following week, therefore laboratory visits ranged from a minimum of 2 to a maximum of 4 sessions per week. This amounted to 24 training visits consisting of aerobic exercise training and water immersion within the required 8-week period.

Tables. Please highlight any significant effects to make it easy for the reader.

Significant effects have now been highlighted bold in all tables.

Tables and figures. Please add the statistical test that was performed and effect sizes.

Statistical tests have been added to figure legends and effect sizes have been provided throughout the manuscript.

Referee #2:

Overview

The objective of this study was to determine if post-exercise hot water immersion improves cardiorespiratory fitness, vascular health and metabolic health to a greater extent than post-exercise thermoneutral water immersion (lines 138-140). Relatively healthy middle-aged adults were recruited and randomized to 8 weeks of aerobic exercise training followed by immersion in 34 or 40 {degree sign}C water. The outcome variables were measured before and after the intervention. The results show a greater pre/post change in diastolic blood pressure, mean arterial pressure, and flow-mediated dilation in the hot water immersion group (relative to the change in the thermoneutral group). There were no differences in the pre/post change between groups for the other variables. The authors conclude that "8-weeks of post-exercise hot water immersion further improves diastolic blood pressure, mean arterial blood pressure and brachial artery endothelial function. These vascular and haemodynamic benefits were apparent in the absence of any additional improvements in other traditional CVD risk factors, such as cardiorespiratory fitness, circulating lipids and inflammatory markers in physically inactive populations that do not achieve the minimum recommended physical activity guidelines." (lines 754-760).

Main comments

1) I congratulate the authors for conducting this challenging, comprehensive study, and

original study. However, I have several concerns regarding the reporting of the study that dampen my enthusiasm. Given this was labelled as a clinical trial, I encourage the authors provide a more transparent reporting of the methods and results. Specifically:

We thank the reviewer for their positive comments regarding the study. We acknowledge that the clinical trials registration was not given the appropriate oversight when it was originally completed, and we should have kept it up to date more effectively. We have responded to each point individually raised below and have been open and detailed in our responses throughout. We consider prior registration an important part of transparent science and we will strive to improve our approach in future work.

1.1. The manuscript would benefit from a more explicit presentation of primary/secondary/exploratory outcomes. The presentation of the methods, the results and discussion should then be aligned with the importance that was given to each variable. I note that the CT registration lists 25 primary outcomes! Was the study powered to detect the stated effect size for each of these variables? How did the analyses consider multiplicity of testing for so many primary outcomes (no correction is stated in the analysis section of the manuscript).

Following the reviewer's comment, we have now explicitly outlined what we considered as primary, secondary and exploratory outcome measures in the methods.

L 219-223: Alongside the primary outcome measures of SBP, DBP and MAP, this study also assessed secondary outcome measures, including brachial artery flow-mediated dilation, carotid to femoral pulse wave velocity, and cardiorespiratory fitness. Additionally, exploratory outcome measures included glucose tolerance, venous lipids, inflammatory, angiogenic and vasoactive markers, future cardiovascular disease risk, and quality of life.

The order of our discussion has remained the same as it is already in line with the importance of our outcome measures. In addition, we have not changed the order of the methods section as we believe that this is better reported in the order in which measurements were taken. However, if the reviewer feels strongly about this, we are open to changing it.

It was not intended that the study have 25 primary outcome variables. Upon reflection, the clinical trials registration was not given the appropriate oversight when it was originally completed, and this is simply an error. The study was powered on blood pressure (SBP, DBP and MAP) as the primary outcome variables and the stated sample size on the clinical trials registration matches this a priori calculation.

1.2. There are several inconsistencies between what is reported on the CT registration and what is reported in the manuscript:

- Lines 185-188: was randomization performed before the pre-intervention assessments? Was randomization performed by someone not involved in the study? Was the sequence known by the study staff? The CT registration states triple blinding; please explain how this was ensured, especially for study participants as I don't see how they could be blinded to hot vs thermoneutral water immersion.

The statement of triple blinding in the clinical trials registration is not correct. Further information has been provided in the manuscript to explain how and when the randomisation was performed.

As correctly stated by the reviewer, the participants were not blinded to hot and thermoneutral water immersion, rather participants were not informed which was the 'active' and which was the 'sham' arm of the intervention.

L 189-194: Randomisation took place before pre intervention health assessments by the principal investigator. Participants were not informed that the EX+HWI group was the 'active' arm of the intervention, while the EX+TWI group served as the 'sham' group. As the principal investigator had to monitor water temperature and be aware of the risk of pre-syncope symptoms (i.e. dizziness and nausea) when exiting the hot tub, the principal investigator was not blinded to group allocation.

- Some primary outcomes (e.g., glycated haemoglobin) and secondary outcomes (e.g., core temperature, thermal comfort/sensation, affect, physical activity enjoyment) listed on the CT registration are not presented. This gives an impression of selective bias in the reporting of outcomes.

In the case of glycated haemoglobin and core temperature, these outcomes were not obtained for logistical reasons and unfortunately the trials registry was not updated to reflect this.

We have now included physiological (tympanic temperature and heart rate) and perceptual measures (e.g. physical activity enjoyment, thermal comfort/sensation and affect) to the supplementary data and adjusted the methods accordingly. These measures were not included in the original manuscript as we felt it did not fit the focus of the journal and we felt they were not particularly novel, but upon reflection feel that including this data (at least as supplementary data) may help researchers and practitioners better understand the stimulus and people's acute responses within this particular study.

L 415-425: Characterisation of the heating stimulus

Resting tympanic temperature (Braun ThermoScan, Kronberg, Germany), heart rate (FT1 Polar, Espoo, Finland) and perceptual measures, including thermal sensation (+5 hot to -5 cold), thermal comfort (+5 very comfortable to -5 very uncomfortable) (Gagge *et al.*, 1967), and basic affect (+5 very good to -5 very bad) (Williams *et al.*, 2008) were recorded before

and after the first and final exercise plus immersion sessions for each intervention. Participants also completed an 8-item Physical Activity Enjoyment Scale after the first and final sessions, which included subdomains of pleasure, fun, pleasant, invigorating, gratifying, exhilarating, stimulating and refreshing (7 positive to 1 negative) (Mullen *et al.*, 2011). These data were used to characterise basic psychological and physiological responses to the acute heating stimulus and can be found in the supplementary data.

- Some protocol deviations may have occurred and are not presented. For example, the CT registration lists venous blood samples for glycaemic variables but the manuscript states capillary measurements. The age range for inclusion is stated as 45-60 but a range of 45-65 is stated on the CT registration. Lastly, the SF-36 questionnaire is not listed on the CT registration but is presented in the manuscript.

The reviewer is correct that blood glucose was measured from capillary blood, and we apologise that this detail was not updated in the clinical trials registration. We also apologise for the omission of the SF-36, again this is a case of not updating the registration.

We believe the second point about age ranges is not accurate. The wording in our manuscript simply describes the age range of participants in the study, not what our inclusion criteria was, as the reviewer suggests. However, to make the inclusion and exclusion criteria clearer in the manuscript, we have re-worded this section to provide clarity on this point.

L 172-183: The inclusion criteria for this study required participants to be aged between 45-60 years and classified as physically inactive, defined as engaging in <150 minutes of moderate intensity or <75 minutes of vigorous intensity activity per week, over the 6-month period prior to the study. Only non-smokers and individuals not taking anti-inflammatory medications were recruited. Self-confirmed postmenopausal women (i.e. 12 months without menstruation), who were not on hormone replacement therapy, were recruited to minimise the effect on circulating hormones on cardiovascular responses (Minson *et al.*, 2000). Participants were excluded if they regularly used heat therapy (e.g. sauna and hot water immersion) or had travelled to a warm country in the 4-weeks before commencing the study. Participants were also excluded if they had a history of orthostatic hypotension, as passive heating can exacerbate pre-syncope symptoms (Steward *et al.*, 2023) or if they had cardiovascular, metabolic or renal disease.

1.3. The objective of the study, as stated on the CT registration and in the manuscript, was to determine if post-exercise hot water immersion provides a "greater" effect than post-exercise thermoneutral water immersion. Accordingly, the analyses, results and interpretation/discussion should be restricted to the between-intervention differences in change scores.

We appreciate the reviewer's point that between-group differences align with the primary objective of the study as registered and stated in the manuscript. However, we respectfully ask the reviewer to reconsider the value of our secondary within-group analyses. We believe the inclusion of within-group differences provide important context for interpretation and practical application.

Given our active-comparator design, where both interventions (EX+HWI and EX+TWI) were expected to potentially elicit improvements, we believe that within-group analyses are justified. They offer useful insights into the direction and magnitude of change within each intervention arm, which may be particularly informative in clinical and applied settings. One good example of this being that 30 min of moderate intensity exercise, in line with the minimum recommended guidelines, did not improve many of the health parameters when followed by TWI.

Whilst the primary objective was to assess between-group differences, omitting within-group changes could limit the translational value of our findings. These data help practitioners and researchers understand how each intervention performed independently, which we believe complements the between-group differences. In this regard, excluding within-group results could give the appearance of selective reporting, whereas including ensures transparency and completeness.

Finally, we wish to clarify that within-group comparisons are presented solely as secondary, exploratory outcomes. This is explicitly stated in the manuscript, and we have been careful not to draw firm conclusions based on within-group results alone.

L 448-452: As secondary analysis, paired t-tests, Wilcoxon tests or Mann-Whitney U tests were used to assess the within-group responses to each intervention. Within-group testing enabled us to establish the significance and magnitude of any changes in response to each individual intervention separately, providing clinically relevant insights.

We hope the reviewer will consider this perspective in their next round of comments. We are open to further discussion, if the reviewer feels strongly about this point.

2) Lines 195-197: I appreciate the inclusion of the expected effect size, but it would be important to specify the absolute difference in blood pressure that was expected and/or that was considered important before starting the study. It is also unclear why this effect size is only presented for blood pressure given the uncertainty regarding the primary outcomes and it is also unclear if one measure of blood pressure (systolic, diastolic, mean) was considered more important than others before undertaking the study.

In response to the reviewer's comment, we have now provided the absolute differences in blood pressure alongside effect sizes. The study was powered to blood pressure (SBP, DBP and MAP) as the primary outcome variables.

L 202-223: An a priori sample size calculation ($\alpha = 0.05$ and $\beta = 0.80$) determined a total of 24 participants (12 in each group) would be required to detect an effect size of $d = 0.3$ using a repeated measures design (within-between interaction), with 2 groups and 2 measurement time points for our primary outcome measures of systolic blood pressure (SBP), diastolic blood pressure (DBP) and mean arterial blood pressure (MAP). This effect size was chosen based on the study of Akerman et al., (2019), who reported meaningful absolute changes, which equated to moderate to large effect sizes for reductions in SBP (-4 mmHg, $d = 0.57$), DBP (-3 mmHg, $d = 0.43$) and MAP (-3 mmHg, $d = 0.50$) following 12 weeks of pre-exercise heat therapy compared to exercise alone. This is supported by previous research indicating that lowering DBP by 2 mmHg in middle-aged men and women with a DBP >90 mmHg reduces the risk of coronary heart disease by 6% and stroke by 15% (Cook, 1995). Additionally, reducing SBP by 5 mmHg lowers the risk of future CVD events by 10% in both men and women around the age of 65, regardless of their initial blood pressure levels (Rahimi *et al.*, 2021). Taken together, based on the study by Akerman and colleagues, we chose to use a more conservative estimate ($d = 0.3$) of their observed blood pressure changes given the different populations being studied and protocols being used.

Alongside the primary outcome measures of SBP, DBP and MAP, this study also assessed secondary outcome measures, including brachial artery flow-mediated dilation, carotid to femoral pulse wave velocity, and cardiorespiratory fitness. Additionally, exploratory outcome measures included glucose tolerance, venous lipids, inflammatory, angiogenic and vasoactive markers, future cardiovascular disease risk, and quality of life.

3) By convention, the participant flowchart, the participant characteristics at baseline (please include in main manuscript), the adherence, and any adverse events (currently not stated) should be presented at the start of the results section for a clinical trial.

The participant flowchart, participant characteristics at baseline, adherence, and adverse events have now been included at the start of the results section as requested by the reviewer.

4) Please include a specific limitations section. I think the most important limitation being that it remains unknown how the observed differences translate into any meaningful effect on longer-term cardiovascular risk/health.

We agree that the manuscript would benefit from a specific limitation paragraph and have added the following paragraph.

L 800-811: Limitations

This study has several limitations that should be considered when interpreting our results. First, longer-term studies are required to establish whether the observed changes in cardiovascular health markers translate into a reduction in CVD incidence. While we observed significant improvements in DBP, MAP, and brachial artery FMD in the EX+HWI compared to the EX+TWI group, the current study was not powered to detect effects in all

secondary and exploratory outcome measures. In addition, it remains unclear whether the improvements in brachial artery blood pressure and endothelial function would translate to other healthy and clinical populations. In this regard, further studies are necessary to establish the ideal dose and mode of post-exercise heat therapy to enhance not only overall health, but also specific cardiovascular and cardiometabolic parameters related to the pathophysiology of different chronic diseases.

Specific comments

5) Lines 132-135: this sentence is not entirely true since a previous study is available (PMID: 35785965). This study is cited in the discussion, but it would be important to present it in the introduction and briefly explain how the current study differs.

We agree with the reviewer that it would also be worth mentioning this study in the introduction.

L 130-137: To date, the only randomised controlled trial investigating the additive health benefits of post-exercise heat therapy was conducted by Lee et al. (2022), who reported greater improvements in cardiorespiratory fitness, systolic blood pressure, and total cholesterol after eight weeks of post-exercise sauna use in physically inactive middle-aged adults. However, it remains unclear whether other forms of heat therapy can produce similar benefits. Moreover, whilst Lee and colleagues assessed several clinically relevant outcomes, they did not measure vascular function, glycaemic control, lipid profiles, and markers of chronic low-grade inflammation.

6) Lines 329-332: blinding was performed for FMD measurements, but what about the other outcomes?

FMD measurements were not blinded, rather the FMD analysis was blinded. The PI was not blinded to group allocation for other outcome measures in this study. This was for a couple of reasons which are described in the manuscript and outlined below.

L 189-194: Randomisation took place before pre intervention health assessments by the principal investigator. Participants were not informed that the EX+HWI group was the 'active' arm of the intervention, while the EX+TWI group served as the 'sham' group. As the principal investigator had to monitor water temperature and be aware of the risk of pre-syncope symptoms (i.e. dizziness and nausea) when exiting the hot tub, the principal investigator was not blinded to group allocation.

7) Line 450: I believe journal policy recommends reporting exact p values when values are greater than 0.05?

Exact p values provided throughout manuscript.

Dear Dr Steward,

Re: JP-RP-2025-288873R1 "Post-exercise hot water immersion enhances vascular benefits of exercise without further improving cardiorespiratory fitness, lipids, or inflammation" by Charles James Steward, Mathew Hill, Campbell Menzies, Sophie Lauren Russell, C Doug Thake, Christopher Pugh, and Tom Cullen

Thank you for submitting your manuscript to The Journal of Physiology. It has been assessed by a Reviewing Editor and by 3 expert referees and we are pleased to tell you that it is acceptable for publication following satisfactory revision.

REVISION CHECKLIST:

We look forward to receiving your revised submission.

Yours sincerely,

Karyn Hamilton
Senior Editor
The Journal of Physiology

REQUIRED ITEMS

- The contact information for the person responsible for 'Research Governance' at your institution needs to be provided. This includes their name and an institutional email address. Please ensure the contact is not an author on this paper and provide an alternate contact if necessary, or confirm in the submission form that the author whose email was provided has sole responsibility for research governance. This is the person who is responsible for regulations, principles and standards of good practice in research carried out at the institution, for instance the ethical treatment of animals, the keeping of proper experimental records or the reporting of results.

- Please remove the BioRender logo from your Abstract Figure. To do this, use The Journal's BioRender account. Follow this link and enter your details and the manuscript number to create and download figures. Upload these as the figure files for your revised submission. If you choose not to take up this offer, we require figures to be of similar quality and resolution. If you are opting out of this service to authors, state this in the Comments section on the Detailed Information page of the submission form. The link provided should only be used for the purposes of this submission. Authors will be charged for figures created on this premium BioRender account if they are not related to this manuscript submission.

EDITOR COMMENTS

Reviewing Editor:

Comments to the Author:

Thank you for nicely addressing the comments of the reviewers. There remains one additional comment by Reviewer #2. I have recommended statistical review to come to a consensus on this point. Nevertheless, I encourage the authors to thoughtfully consider this remaining comment. Thanks!

Senior Editor:

Comments for Authors to ensure the paper complies with the Statistics Policy:

Thank you for including precise p-values in the revised figures. Please confirm in Table 1 (in the legend) that variance is represented as SD.

Comments to the Author:

Thank you for submitting your carefully revised manuscript. In large part, your revisions addressed the Referee concerns and, at this point, we would like to Provisionally Accept your manuscript pending a few remaining points to be addressed by the authors. We recruited a statistics expert to provide input on the point raised regarding the need to statistically compare the within-intervention changes, in addition to the between-intervention changes. We look forward to receiving your revised manuscript and appreciate your interest in The Journal.

REFEREE COMMENTS

Referee #1:

The authors have done a great job addressing all of my previous comments. Thank you.

Referee #2:

Thank you for considering my comments. I appreciate the addition of important details that improve the transparency of the study. I am still not convinced that the within intervention changes require statistical testing. The authors argue that this approach provides clinical insights into the effect of each intervention. I can agree with this, but statistical testing is not needed to achieve this objective. The authors could simply report the mean difference with confidence interval for each intervention (without statistical testing) and then present the statistical comparison between interventions. I am mostly concerned about the multiplicity of statistical testing that does not seem to have been accounted for. Perhaps I am wrong, and I would be happy to defer to the editor or a statistical reviewer.

Also, please check line 147 as I believe it should read "maximal oxygen uptake" rather than "submaximal oxygen uptake"?

And please confirm that the between-group difference for change in the mental health component score for the quality of life questionnaire was 4 (line 636). I am asking because the change within Ex+HWI the change was 10 units (line 638) compared to 1 unit within Ex+TWI (line 641) which gives a difference of 9 units, not 4.

Referee #3:

Comments for Author:

I agree with the authors that the within-group analyses are justified, as they offer useful insights into the direction and magnitude of changes within each intervention arm, which is particularly relevant given the expectation that both interventions could elicit improvements. Similarly, I share the reviewer's concerns that these may be over-interpreted and subject to inflated type I error rates due to the multiple testing. As such, I support the reviewer's recommendation that the authors retain the within-group changes, but just report the mean difference with confidence intervals for each intervention (without statistical testing). This position balances methodological soundness with clinical/practical relevance. The authors should explicitly state this approach in the manuscript (e.g., "For exploratory within-group analyses, we report descriptive statistics and 95% CIs to aid interpretation, omitting p-values to avoid issues with multiple comparisons").

END OF COMMENTS

EDITOR COMMENTS

Reviewing Editor:

Comments to the Author:

Thank you for nicely addressing the comments of the reviewers. There remains one additional comment by Reviewer #2. I have recommended statistical review to come to a consensus on this point. Nevertheless, I encourage the authors to thoughtfully consider this remaining comment. Thanks!

We would like to thank the Reviewing Editor for taking the time to handle our manuscript.

Senior Editor:

Comments for Authors to ensure the paper complies with the Statistics Policy:

Thank you for including precise p-values in the revised figures. Please confirm in Table 1 (in the legend) that variance is represented as SD.

Data are presented as means \pm SD. This has now in the legend of Table 1.

Comments to the Author:

Thank you for submitting your carefully revised manuscript. In large part, your revisions addressed the Referee concerns and, at this point, we would like to Provisionally Accept your manuscript pending a few remaining points to be addressed by the authors. We recruited a statistics expert to provide input on the point raised regarding the need to statistically compare the within-intervention changes, in addition to the between-intervention changes. We look forward to receiving your revised manuscript and appreciate your interest in The Journal.

We would like to thank the Senior Editor for their oversight throughout the review process.

REFEREE COMMENTS

Referee #1:

The authors have done a great job addressing all of my previous comments. Thank you.

We would like to thank Reviewer 1 for taking the time to review our manuscript and for providing valuable feedback. We sincerely appreciate your constructive comments,

which have helped us to improve the quality and clarity of our study. Your insights have contributed meaningfully to refining our work, and we are grateful for your support in strengthening our manuscript.

Referee #2:

Thank you for considering my comments. I appreciate the addition of important details that improve the transparency of the study. I am still not convinced that the within intervention changes require statistical testing. The authors argue that this approach provides clinical insights into the effect of each intervention. I can agree with this, but statistical testing is not needed to achieve this objective. The authors could simply report the mean difference with confidence interval for each intervention (without statistical testing) and then present the statistical comparison between interventions. I am mostly concerned about the multiplicity of statistical testing that does not seem to have been accounted for. Perhaps I am wrong, and I would be happy to defer to the editor or a statistical reviewer.

Thank you for your thoughtful follow-up. We appreciate your perspective, and after careful consideration, we agree that omitting statistical testing for our exploratory within-group analyses is the appropriate course of action given concerns regarding the multiplicity of statistical testing. We have now removed all p-values from our within-group comparisons and have presented only descriptive statistics and 95% confidence intervals.

In addition, we have also removed or amended any text that referred to the within-group statistical analyses.

L 681-683: However, we observed meaningful reductions in the majority of blood pressure parameters within the post-exercise hot water immersion group, which were not apparent following post-exercise thermoneutral water immersion.

Also, please check line 147 as I believe it should read "maximal oxygen uptake" rather than "submaximal oxygen uptake"?

This has now been corrected to maximal oxygen uptake.

And please confirm that the between-group difference for change in the mental health component score for the quality of life questionnaire was 4 (line 636). I am asking because the change within Ex+HWI the change was 10 units (line 638) compared to 1 unit within Ex+TWI (line 641) which gives a difference of 9 units, not 4.

We can confirm that the between-group difference for the Quade's ANCOVA is correct. There was a small error for the initial within-group analysis data set for this particular outcome measure. We would like to thank the reviewer for spotting this mistake, this

has now been changed in the manuscript.

Referee #3:

Comments for Author:

I agree with the authors that the within-group analyses are justified, as they offer useful insights into the direction and magnitude of changes within each intervention arm, which is particularly relevant given the expectation that both interventions could elicit improvements. Similarly, I share the reviewer's concerns that these may be over-interpreted and subject to inflated type I error rates due to the multiple testing. As such, I support the reviewer's recommendation that the authors retain the within-group changes, but just report the mean difference with confidence intervals for each intervention (without statistical testing). This position balances methodological soundness with clinical/practical relevance. The authors should explicitly state this approach in the manuscript (e.g., "For exploratory within-group analyses, we report descriptive statistics and 95% CIs to aid interpretation, omitting p-values to avoid issues with multiple comparisons").

We agree with the combined responses of reviewer 2 and 3. We have now removed all within-group p values and have only provided descriptive statistics and 95% confidence intervals throughout the manuscript.

L 458-465: To assist with data interpretation, mean and confidence intervals [95% CI] were calculated for parametric data, whilst median with quartile 1 (25th percentile) and 3 (75th percentile) [Q1, Q3] were provided for non-parametric data. In addition, effect sizes were calculated as partial eta squared (η^2_p) and Hedges' g to assist with assessing the practical significance of findings. This approach also enabled us to establish the magnitude of any changes in response to each individual intervention separately, providing clinically relevant insights. For exploratory within-group analyses P -values to avoid issues with multiple comparisons.

Dear Dr Steward,

Re: JP-RP-2025-288873R2 "Post-exercise hot water immersion enhances vascular benefits of exercise without further improving cardiorespiratory fitness, lipids, or inflammation" by Charles James Steward, Mathew Hill, Campbell Menzies, Sophie Lauren Russell, C Doug Thake, Christopher Pugh, and Tom Cullen

Thank you for submitting your manuscript to The Journal of Physiology. It has been assessed by a Reviewing Editor and we are pleased to tell you that it is acceptable for publication following satisfactory revision.

REVISION CHECKLIST:

We look forward to receiving your revised submission.

Yours sincerely,

Karyn Hamilton
Senior Editor
The Journal of Physiology

EDITOR COMMENTS

Reviewing Editor:

Comments to the Author:

Thank you for nicely addressing the comments of the reviewers. I have found a typo, but otherwise looks good. Thank you for your submission to J Physiol!

Minor typo:

Ln 474-475 - It seems like "have been omitted" should be added to this sentence. E.g., "... within-group analyses P-values [have been omitted] to avoid issues with..."

Senior Editor:

Comments to the Author:

Thank you for these revisions. At this point, the Referees are pleased with the revisions. There's is just one potential typographical error (see the Reviewing Editor's note) to check on before we can accept your manuscript for publication. It should be a quick fix!

EDITOR COMMENTS

Reviewing Editor:

Comments to the Author:

Thank you for nicely addressing the comments of the reviewers. There remains one additional comment by Reviewer #2. I have recommended statistical review to come to a consensus on this point. Nevertheless, I encourage the authors to thoughtfully consider this remaining comment. Thanks!

We would like to thank the Reviewing Editor for taking the time to handle our manuscript.

Senior Editor:

Comments for Authors to ensure the paper complies with the Statistics Policy:

Thank you for including precise p-values in the revised figures. Please confirm in Table 1 (in the legend) that variance is represented as SD.

Data are presented as means \pm SD. This has now in the legend of Table 1.

Comments to the Author:

Thank you for submitting your carefully revised manuscript. In large part, your revisions addressed the Referee concerns and, at this point, we would like to Provisionally Accept your manuscript pending a few remaining points to be addressed by the authors. We recruited a statistics expert to provide input on the point raised regarding the need to statistically compare the within-intervention changes, in addition to the between-intervention changes. We look forward to receiving your revised manuscript and appreciate your interest in The Journal.

We would like to thank the Senior Editor for their oversight throughout the review process.

REFEREE COMMENTS

Referee #1:

The authors have done a great job addressing all of my previous comments. Thank you.

We would like to thank Reviewer 1 for taking the time to review our manuscript and for providing valuable feedback. We sincerely appreciate your constructive comments,

which have helped us to improve the quality and clarity of our study. Your insights have contributed meaningfully to refining our work, and we are grateful for your support in strengthening our manuscript.

Referee #2:

Thank you for considering my comments. I appreciate the addition of important details that improve the transparency of the study. I am still not convinced that the within intervention changes require statistical testing. The authors argue that this approach provides clinical insights into the effect of each intervention. I can agree with this, but statistical testing is not needed to achieve this objective. The authors could simply report the mean difference with confidence interval for each intervention (without statistical testing) and then present the statistical comparison between interventions. I am mostly concerned about the multiplicity of statistical testing that does not seem to have been accounted for. Perhaps I am wrong, and I would be happy to defer to the editor or a statistical reviewer.

Thank you for your thoughtful follow-up. We appreciate your perspective, and after careful consideration, we agree that omitting statistical testing for our exploratory within-group analyses is the appropriate course of action given concerns regarding the multiplicity of statistical testing. We have now removed all p-values from our within-group comparisons and have presented only descriptive statistics and 95% confidence intervals.

In addition, we have also removed or amended any text that referred to the within-group statistical analyses.

L 681-683: However, we observed meaningful reductions in the majority of blood pressure parameters within the post-exercise hot water immersion group, which were not apparent following post-exercise thermoneutral water immersion.

Also, please check line 147 as I believe it should read "maximal oxygen uptake" rather than "submaximal oxygen uptake"?

This has now been corrected to maximal oxygen uptake.

And please confirm that the between-group difference for change in the mental health component score for the quality of life questionnaire was 4 (line 636). I am asking because the change within Ex+HWI the change was 10 units (line 638) compared to 1 unit within Ex+TWI (line 641) which gives a difference of 9 units, not 4.

We can confirm that the between-group difference for the Quade's ANCOVA is correct. There was a small error for the initial within-group analysis data set for this particular outcome measure. We would like to thank the reviewer for spotting this mistake, this

has now been changed in the manuscript.

Referee #3:

Comments for Author:

I agree with the authors that the within-group analyses are justified, as they offer useful insights into the direction and magnitude of changes within each intervention arm, which is particularly relevant given the expectation that both interventions could elicit improvements. Similarly, I share the reviewer's concerns that these may be over-interpreted and subject to inflated type I error rates due to the multiple testing. As such, I support the reviewer's recommendation that the authors retain the within-group changes, but just report the mean difference with confidence intervals for each intervention (without statistical testing). This position balances methodological soundness with clinical/practical relevance. The authors should explicitly state this approach in the manuscript (e.g., "For exploratory within-group analyses, we report descriptive statistics and 95% CIs to aid interpretation, omitting p-values to avoid issues with multiple comparisons").

We agree with the combined responses of reviewer 2 and 3. We have now removed all within-group p values and have only provided descriptive statistics and 95% confidence intervals throughout the manuscript.

L 458-465: To assist with data interpretation, mean and confidence intervals [95% CI] were calculated for parametric data, whilst median with quartile 1 (25th percentile) and 3 (75th percentile) [Q1, Q3] were provided for non-parametric data. In addition, effect sizes were calculated as partial eta squared (η^2_p) and Hedges' g to assist with assessing the practical significance of findings. This approach also enabled us to establish the magnitude of any changes in response to each individual intervention separately, providing clinically relevant insights. For exploratory within-group analyses, P -values have been omitted to avoid issues with multiple comparisons.

Dear Dr Steward,

Re: JP-RP-2025-288873R3 "Post-exercise hot water immersion enhances vascular benefits of exercise without further improving cardiorespiratory fitness, lipids, or inflammation" by Charles James Steward, Mathew Hill, Campbell Menzies, Sophie Lauren Russell, C Doug Thake, Christopher Pugh, and Tom Cullen

We are pleased to tell you that your paper has been accepted for publication in The Journal of Physiology.

Yours sincerely,

Karyn Hamilton
Senior Editor
The Journal of Physiology

If you would like to receive our 'Research Roundup', a monthly newsletter highlighting the cutting-edge research published in The Physiological Society's family of journals (The Journal of Physiology, Experimental Physiology, Physiological Reports, The Journal of Nutritional Physiology and The Journal of Precision Medicine: Health and Disease), please click this link, fill in your name and email address and select 'Research Roundup':
<https://www.physoc.org/journals-and-media/membernews>

- You can help your research get the attention it deserves! Check out Wiley's free Promotion Guide for best-practice recommendations for promoting your work at: www.wileyauthors.com/eeo/guide. You can learn more about Wiley Editing Services which offers professional video, design, and writing services to create shareable video abstracts, infographics, conference posters, lay summaries, and research news stories for your research at: www.wileyauthors.com/eeo/promotion.

EDITOR COMMENTS

Reviewing Editor:

Comments to the Author:

Congratulations on a very nice study. I agree with the reviewers that this has high potential to have a substantial impact.

Senior Editor:

Comments to the Author:

Thank you for making this minor revision. We are pleased to accept your manuscript for publication in The Journal of Physiology. Thank you for your interest in The Journal!